# Fully Dynamic $k$-Clustering in $\tilde{O}(k)$ Update Time

**Sayan Bhattacharya**
University of Warwick
s.bhattacharya@warwick.ac.uk

**Martín Costa**
University of Warwick
martin.costa@warwick.ac.uk

**Silvio Lattanzi**
Google Research
silviol@google.com

**Nikos Parotsidis**
Google Research
nikosp@google.com

## Abstract

We present a $O(1)$-approximate fully dynamic algorithm for the $k$-median and $k$-means problems on metric spaces with amortized update time $\tilde{O}(k)$ and worst-case query time $\tilde{O}(k^2)$. We complement our theoretical analysis with the first in-depth experimental study for the dynamic $k$-median problem on general metrics, focusing on comparing our dynamic algorithm to the current state-of-the-art by Henzinger and Kale [20]. Finally, we also provide a lower bound for dynamic $k$-median which shows that any $O(1)$-approximate algorithm with $\tilde{O}(\text{poly}(k))$ query time must have $\tilde{\Omega}(k)$ amortized update time, even in the incremental setting.

## 1 Introduction

Clustering is a fundamental problem in unsupervised learning with several practical applications. In clustering, one is interested in partitioning elements into different groups (i.e. *clusters*), so that elements in the same group are more similar to each other than to elements in other groups. One of the most studied formulations of clustering is the metric clustering formulation. In this setting, elements are represented by points in a metric space, and the distances between points represent how similar the corresponding elements are (the closer elements are, the more similar they are). More formally, the input to our problem consists of a set of points $U$ in a metric space with distance function $d : U \times U \to \mathbb{R}_{\geq 0}$, a real number $p \geq 1$, and an integer $k \geq 1$. The goal is to compute a subset $S \subseteq U$ of size $|S| \leq k$, so as to minimize $\texttt{cost}(S) := \sum_{x \in U} d(x, S)^p$, where $d(x, S) := \min_{y \in S} d(x, y)$. We refer to the points in $S$ as *centers*. We note that this captures well-known problems such as $k$-median clustering (when $p = 1$) or $k$-means clustering (when $p = 2$).

Due to this simple and elegant formulation, metric clustering has been extensively studied throughout the years, across a range of computational models [14, 22, 1, 12, 15, 2, 31, 11, 7]. In this paper, we focus on the *dynamic setting*, where the goal is to efficiently maintain a good clustering when the input data keeps changing over time. Because of its immediate real-world applications, the dynamic clustering problem has received a lot of attention from the machine learning community in recent years [16, 6, 10, 24, 18]. Below, we formally describe the model [13, 20, 3] considered in this paper.

At *preprocessing*, the algorithm receives the initial input $U$. Subsequently, the input keeps changing by means of a sequence of *update* operations, where each update inserts/deletes a point in $U$. Throughout this sequence of updates, the algorithm needs to implicitly maintain a solution $S^* \subseteq U$ to the current input $(U, d)$. The algorithm has an approximation ratio of $\rho \geq 1$ if and only if we always have $\texttt{cost}(S^*) \leq \rho \cdot \texttt{opt}(U)$, where $\texttt{opt}(U) := \min_{S \subseteq U, |S| \leq k} \texttt{cost}(S)$ denotes the optimal objective. Finally, whenever one *queries* the algorithm, it has to return the list of centers in $S^*$. The performance of a dynamic algorithm is captured by its approximation ratio, its update time, and its query time. Let

37th Conference on Neural Information Processing Systems (NeurIPS 2023).

$U_{\text{INIT}}$ be the state of the input $U$ at preprocessing. A dynamic algorithm has (amortized) *update time* $O(\lambda)$ if for any sequence of $t \geq 0$ updates, the total time it spends on preprocessing and handling the updates is $O((t + |U_{\text{INIT}}|) \cdot \lambda)$. The time it takes to answer a query is called its *query time*.

**Our Contributions.** Our primary contribution is to design an algorithm for this problem with a near-optimal update time of $\tilde{O}(k)$, without significantly compromising on the approximation ratio and query time.[1] Interestingly, our algorithm can be easily extended to dynamic $k$-clustering for any constant $p$ (see Appendix A). In the theorem below, we summarize our main result for dynamic $k$-median and $k$-means.

**Theorem 1.1.** *There exists a dynamic algorithm that, with high probability, maintains a $O(1)$-approximate solution to the $k$-median and $k$-means problems for general metric spaces under point insertions and deletions with $\tilde{O}(k)$ amortized update time and $\tilde{O}(k^2)$ worst-case query time.*

It is important to note that in practice an algorithm often receives the updates in *batches* of variable sizes [25, 29], so it is common for there to be a significant number of updates between cluster requests. As a consequence, optimizing the update time is of primary importance in practical applications. For example, if the size of each batch of updates is $\Omega(k)$, then our amortized query time becomes $\tilde{O}(k^2/k) = \tilde{O}(k)$, because the algorithm has to answer a query after processing at least one batch. Thus, our dynamic algorithm has near-optimal update and query times when the updates arrive in moderately large batches.

In addition, it is possible to observe that any dynamic $k$-clustering algorithm must have $\Omega(k)$ query time, since the solution it needs to return while answering a query can consist of $k$ centers. We also show a similar lower bound on the *update time* of any constant approximate dynamic algorithm for this problem. This lower bound holds even in an *incremental* setting, where we only allow for point-insertions in $U$. We defer the proof of Theorem 1.2 to Appendix D.

**Theorem 1.2.** *Any $O(1)$-approximate incremental algorithm for the $k$-median problem with $\tilde{O}(\text{poly}(k))$ query time must have $\tilde{\Omega}(k)$ amortized update time.*

Theorem 1.2 implies that the ultimate goal in this line of research is to get a $O(1)$-approximate dynamic $k$-clustering algorithm with $\tilde{O}(k)$ update time and $\tilde{O}(k)$ query time. Prior to this work, however, the state-of-the-art result for dynamic $k$-median (and $k$-means) was due to Henzinger and Kale [20], who obtained $O(1)$-approximation ratio, $\tilde{O}(k^2)$ update time and $\tilde{O}(k^2)$ query time. In this context, our result is a meaningful step toward obtaining an asymptotically optimal algorithm for the problem.

We supplement the above theorem with experiments that compare our algorithm with that of [20]. To the best of our knowledge, this is the first work in the dynamic clustering literature with a detailed experimental evaluation of dynamic $k$-median algorithms for general metrics. Interestingly, we observe that experimentally our algorithm is significantly more efficient than previous work without impacting the quality of the solution.

**Our Techniques.** We first summarize the approach used in the previous state-of-the-art result. In [20], the authors used a static algorithm for computing a coreset of size $\tilde{O}(k)$ as a black box to get a fully dynamic algorithm for maintaining a coreset of size $\tilde{O}(k)$ for general metric spaces in worst-case update time $\tilde{O}(k^2)$. Their algorithm works by maintaining a balanced binary tree of depth $O(\log n)$, where each leaf in the tree corresponds to a point in the metric space [4]. Each internal node of the tree then takes the union of the (weighted) sets of points maintained by its two children and computes its coreset, which is then fed to its parent. They maintain this tree dynamically by taking all the leaves affected by an update and recomputing all the coresets at nodes contained in their leaf-to-root paths. Using state-of-the-art static coreset constructions, this update procedure takes time $\tilde{O}(k^2)$. Unfortunately, if one wants to ensure that the final output is a valid corset of the metric space after each update, then there is no natural way to bypass having to recompute the leaf-to-root path after the deletion of a point that is contained in the final coreset. Hence, it is not at all clear how to modify this algorithm in order to reduce the update time to $\tilde{O}(k)$.

---

[1]Throughout this paper, we use $\tilde{O}(.)$ and $\tilde{\Omega}(.)$ notations to suppress $\text{polylog}(n)$ factors, where $n = |U|$.

We circumvent this bottleneck by taking a completely different approach compared to [20]. Our algorithm instead follows from the dynamization of a $\tilde{O}(nk)$ time static algorithm for $k$-median by Mettu and Plaxton [28], where $n = |U|$. We refer to this static algorithm as the MP algorithm. Informally, the MP algorithm works by constructing a set of $O(\log(n/k))$ *layers* by iteratively sampling random points at each layer, defining a clustering of the points in the layer that are 'close' to these samples, and removing these clusters from the layer. The algorithm then obtains a solution to the $k$-clustering problem by running a static algorithm for *weighted $k$-median* (defined in Section 2), on an instance of size $O(k \log(n/k))$ defined by the clusterings at each layer. In order to dynamize this algorithm, we design a data structure that maintains $O(\log(n/k))$ layers that are analogous to the layers in the MP algorithm. By allowing for some 'slack' in the way that these layers are defined, we ensure that they can be periodically reconstructed in a way that leads to good amortized update time, while only incurring a small loss in the approximation ratio. The clustering at each layer is obtained by random sampling every time it is reconstructed and is maintained by arbitrarily reassigning a point in a cluster as the new center when the current center is deleted. We obtain a solution to the $k$-clustering problem from this data structure in the same way as the (static) MP algorithm—by running a static algorithm for weighted $k$-median on an instance defined by the clusterings maintained at each layer.

## 2 Preliminaries

In our computational model, the algorithm has access to the distance $d(x, y)$ for any $x, y \in U$ in $O(1)$ time[2]. Given any two sets $X, S \subseteq U$, define *the cost of $X$ w.r.t. $S$* as $\mathtt{cost}(S, X) := \sum_{x \in X} \min_{s \in S} d(x, s)$. In addition, let $\mathtt{cost}(S) = \mathtt{cost}(S, U)$. Next, define an *assignment* to be a function $\pi : U \to U$, and say that $\pi$ assigns $x \in U$ to $\pi(x)$. We refer to an assignment $\pi$ with $|\pi(U)| \leq m$ as an *$m$-assignment*. The cost of $X \subseteq U$ w.r.t. assignment $\pi$ is $\mathtt{cost}(\pi, X) := \sum_{x \in X} d(x, \pi(x))$. We denote $\mathtt{cost}(\pi, U)$ by $\mathtt{cost}(\pi)$. For any $\rho \geq 1$, say that $\pi$ is $\rho$-approximate if $\mathtt{cost}(\pi) \leq \rho \cdot \mathtt{opt}(U)$, where $\mathtt{opt}(U)$ is the cost of the optimal solution.

Given any subset $U' \subseteq U$ and $x \in U'$, we denote by $B_{U'}(x, r)$ the set $\{y \in U' \mid d(x, y) \leq r\}$, i.e. the closed ball in $U'$ of radius $r \geq 0$ centered at $x$. When it is clear from the context that $U' = U$, we omit the subscript $U'$ and simply write $B(x, r)$. For $X \subseteq U' \subseteq U$, we define $B_{U'}(X, r)$ to be the union of the balls $B_{U'}(x, r)$ for $x \in X$.

**Definition 2.1.** Given $0 < \rho < 1$ and subsets $X \subseteq U' \subseteq U$, we define the following real numbers:

$$\nu_\rho(X, U') := \min\{r \geq 0 \mid |B_{U'}(X, r)| \geq \rho \cdot |U'|\}, \mu_\rho(U') := \min\{\nu_\rho(Y, U') \mid Y \subseteq U', \ |Y| = k\}.$$

In words, the real number $\nu_\rho(X, U')$ is the smallest radius $r$ such that the closed ball of radius $r$ around $X$ captures at least a $\rho$-proportion of the points in $U'$. The real number $\mu_\rho(U')$ is then defined to be the smallest such radius $\nu_\rho(X, U')$ over all subsets $X \subseteq U'$ of size $k$. Note that $\nu_\rho(X, U')$ and $\mu_\rho(U')$ are both increasing as functions of $\rho$.

Finally, we will sometimes refer to the *weighted $k$-median problem*. An instance of this problem is given by a triple $(U, d, w)$, where $w : U \to \mathbb{R}_{\geq 0}$ assigns a nonnegative *weight* to every point $x \in U$, in a metric space with distance function $d$. Here, the goal is to compute a subset of at most $k$ centers $S \subseteq U$, so as to minimize $\mathtt{cost}_w(S) := \sum_{x \in U} w(u) \cdot d(x, S)$. We let $\mathtt{opt}_w(U) := \min_{S \subseteq U : |S| \leq k} \mathtt{cost}_w(S)$ denote the optimal objective value of this weighted $k$-median instance. We will analogously use the symbol $\mathtt{cost}_w(S, X) := \sum_{x \in X} w(x) \cdot d(x, S)$ to denote the cost of a set of points $X$ w.r.t. $S$ and the weight function $w$.

## 3 Our Algorithm

Throughout this section, we fix the following parameters: $\alpha$, $\beta$, $\epsilon$, $\tau$ and $k'$; where $\alpha \geq 1$ is a sufficiently large constant, $\beta$ is any constant in the interval $(0, 1)$, $\epsilon > 0$ is a sufficiently small constant, $\tau := \epsilon\beta$, and $k' := \max\{k, \log(|U|)\}$.

---

[2]This is a standard and common assumption in clustering settings.

## 3.1 The Static Algorithm of [28]

Our starting point is the static algorithm of [28] for computing a $\tilde{O}(k)$-assignment $\sigma : U \to U$, with $S = \sigma(U)$, such that $\texttt{cost}(\sigma) \leq O(1) \cdot \texttt{opt}(U)$. The relevant pseudocode appears in Algorithm 1 and Algorithm 2.

---

**Algorithm 1** $\texttt{StaticAlgo}(U)$

---

1: $i \leftarrow 1$ and $U_1 \leftarrow U$
2: **while** $|U_i| > \alpha k'$ **do**
3:      $(S_i, C_i, \sigma_i, \nu_i) \leftarrow \texttt{AlmostCover}(U_i)$
4:      $U_{i+1} \leftarrow U_i \setminus C_i$
5:      $i \leftarrow i + 1$
6: **end while**
7: $t \leftarrow i$
8: $S_t \leftarrow U_t$, $C_t \leftarrow S_t$ and $\nu_t \leftarrow 0$
9: Assign each $x \in C_t$ to itself (i.e., $\sigma_t(x) := x \in S_t$)
10: $S \leftarrow \cup_{j \in [t]} S_j$
11: Let $\sigma : U \to S$ be the assignment such that for all $j \in [t]$ and $x \in C_j$, we have $\sigma(x) = \sigma_j(x)$
12: **return** $(S, \sigma, t)$

---

---

**Algorithm 2** $\texttt{AlmostCover}(U')$

---

1: Construct a set $S'$, by sampling $\alpha k'$ points from $U'$ independently and u.a.r. with replacement
2: $\nu' \leftarrow \nu_\beta(S', U')$ and $C' \leftarrow B(S', \nu')$
3: Assign each $x \in C'$ to some $\sigma'(x) \in S'$ such that $d(x, \sigma'(x)) \leq \nu'$
4: **return** $(S', C', \sigma', \nu')$

---

The algorithm runs for $t$ *iterations*. Let $U_i \subseteq U$ denote the set of unassigned points at the start of iteration $i \in [t-1]$ (initially, we have $U_1 = U$). During iteration $i$, the algorithm samples a set of $\alpha k'$ points $S_i$ as centers, uniformly at random from $U_i$. It then identifies the smallest radius $\nu_i$ such that the balls of radius $\nu_i$ around $S_i$ cover at least $\beta$-fraction of the points in $U_i$. Let $C_i \subseteq U_i$ denote the set of points captured by these balls (note that $C_i \supseteq S_i$ and $|C_i| \geq \beta \cdot |U_i|$). The algorithm assigns each $x \in C_i$ to some center $\sigma_i(x) \in S_i$ within a distance of $\nu_i$. It then sets $U_{i+1} \leftarrow U_i \setminus C_i$, and proceeds to the next iteration. In the very last iteration $t$, we have $|U_t| \leq \alpha k'$, and the algorithm sets $S_t := U_t$, $C_t := U_t$, and $\sigma_t(x) := x$ for each $x \in U_t$.

The algorithm returns the assignment $\sigma$ with set of centers $\sigma(U) = S$, where $\sigma$ is simply the union of $\sigma_i$ for all $i \in [t]$.

Fix any $i \in [t-1]$. Since $|C_i| \geq \beta \cdot |U_i|$, we have $|U_{i+1}| \leq (1 - \beta) \cdot |U_i|$. As $\beta$ is a constant, it follows that the algorithm runs for $t = \tilde{O}(1)$ iterations. Since each iteration picks $O(k)$ new centers, we get: $|S| = \tilde{O}(k)$, and hence $\sigma$ is a $\tilde{O}(k)$-assignment. Furthermore, [28] showed that $\texttt{cost}(\sigma) = O(1) \cdot \texttt{opt}(U)$.

## 3.2 Our Dynamic Algorithm

The main idea behind our dynamic algorithm is that it maintains the output $S$ of the static algorithm from Section 3.1 in a lazy manner, by allowing for some small slack at appropriate places. Thus, it maintains $t$ *layers*, where each layer $i \in [t]$ corresponds to iteration $i$ of the static algorithm. In the dynamic setting, the value of $t$ changes over time. Specifically, layer $i \in [t]$ consists of the tuple $(U_i, S_i, C_i, \sigma_i, \nu_i)$, as in Algorithm 1. Whenever a large fraction of points gets inserted or deleted from some layer $j \in [t]$, the dynamic algorithm rebuilds all the layers $i \in [j, t]$ from scratch.

The dynamic algorithm maintains the sets $U_i$, $S_i$ and $C_i$ explicitly, and the assignment $\sigma_i$ implicitly, in a manner which ensures that for all $x \in C_i$ it can return $\sigma_i(x) \in S_i$ in $O(1)$ time. Furthermore, for each layer $i \in [t]$, it explicitly maintains the value $n_i$ (resp. $n_i^*$), which denotes the size of the set $U_i$ at (resp. the number of updates to $U_i$ since) the time it was last rebuilt. We explain this in more detail below. The value $\nu_i$ is needed only for the sake of analysis.

**Algorithm 3** Preprocess($U$)

1: $U_1 \leftarrow U$
2: ConstructFromLayer($1$)

---

**Algorithm 4** ConstructFromLayer($i$)

1: $j \leftarrow i$
2: **while** $|U_j| > \alpha k'$ **do**
3:    $n_j \leftarrow |U_j|$ and $n_j^* \leftarrow 0$
4:    $(S_j, C_j, \sigma_j, \nu_j) \leftarrow$ AlmostCover($U_j$)
5:    $U_{j+1} \leftarrow U_j \setminus C_j$
6:    $j \leftarrow j + 1$
7: **end while**
8: $t \leftarrow j$
9: $S_t \leftarrow U_t$, $C_t \leftarrow S_t$ and $\nu_t \leftarrow 0$
10: Assign each $x \in C_t$ to itself (i.e., $\sigma_t(x) := x \in S_t$)
11: $S \leftarrow \cup_{j \in [t]} S_i$
12: Let $\sigma : U \rightarrow S$ be the assignment such that for all $j \in [t]$ and $x \in C_j$, we have $\sigma(x) = \sigma_j(x)$

---

**Preprocessing:** At preprocessing, we essentially run the static algorithm from Section 3.1 to set the value of $t$ and construct the layers $i \in [t]$. See Algorithm 3 and Algorithm 4. Note that at this stage $n_j^* = 0$ for all layers $j \in [t]$.

---

**Algorithm 5** Insert($x$)

1: **for** $i = 1, ..., t$ **do**
2:    Add $x$ to $U_i$
3:    $n_i^* \leftarrow n_i^* + 1$
4: **end for**
5: Add $x$ to $C_t$ and $S_t$, and set $\sigma_t(x) \leftarrow x$
6: Rebuild

---

**Handling the insertion of a point $x$ in $U$:** We simply add the point $x$ to $U_i$, for each layer $i \in [t]$. Next, in the last layer $t$, we create a new center at point $x$ and assign the point $x$ to itself. Finally, we call the Rebuild subroutine (to be described below). See Algorithm 5.

**Handling the deletion of a point $x$ from $U$:** Let $j \in [t]$ be the last layer (with the largest index) containing the point $x$. We remove $x$ from each layer $i \in [j]$. Next, if $x$ was a center at layer $j$, then we pick any arbitrary point $x' \in \sigma_j^{-1}(x) \setminus \{x\}$ (if such a point exists), make $x'$ a center, and assign every point $y \in \sigma_j^{-1}(x) \setminus \{x\}$ to the newly created center $x'$. Finally, we call the Rebuild subroutine (to be described below). See Algorithm 6.

**The rebuild subroutine:** We say that a layer $i \in [t]$ is *rebuilt* whenever our dynamic algorithm calls ConstructFromLayer($j$) for some $j \leq i$, and that there is an *update* in layer $i$ whenever we add/remove a point in $U_i$. It is easy to see that $n_i^*$ denotes the number of updates in layer $i \in [t]$ since the last time it was rebuilt (see Line 3 in Algorithm 4, Algorithm 5 and Algorithm 6). Next, we observe that Line 6 in Algorithm 5 and Line 16 in Algorithm 6, along with the pseudocode of Algorithm 7, imply the following invariant.

**Invariant 3.1.** $n_i^* \leq \tau n_i$ for all $i \in [t]$. Here, $n_i$ is the size of $U_i$ just after the last rebuild of layer $i$, and $n_i^*$ is the number of updates in layer $i$ since that last rebuild.

Intuitively, the above invariant ensures that the layers maintained by our dynamic algorithm remain *close* to the layers of the static algorithm in Section 3.1. This is crucially exploited in the proofs of Lemma 3.2 (which helps us bound the update and query times) and Lemma 3.3 (which leads to the desired bound on the approximation ratio). We defer the proofs of these two lemmas to Appendix B.

**Lemma 3.2.** *We always have $t = \tilde{O}(1)$, where $t$ denotes the number of layers maintained by our dynamic algorithm.*

---

**Algorithm 6** Delete($x$)

---

1: **for** $i = 1, ..., t$ **do**
2:   **if** $x \in U_i$ **then**
3:     Remove $x$ from $U_i$, and set $n_i^* \leftarrow n_i^* + 1$
4:     **if** $x \in C_i$ **then**
5:       Remove $x$ from $C_i$
6:       **if** $x \in S_i$ **then**
7:         Remove $x$ from $S_i$
8:         **if** $\exists\, y \in \sigma_i^{-1}(x) \setminus \{x\}$ **then**
9:           Pick any such $y$ and place it into $S_i$
10:           Set $\sigma_i(z)$ to $y$ for each $z \in \sigma_i^{-1}(x)$
11:         **end if**
12:       **end if**
13:     **end if**
14:   **end if**
15: **end for**
16: Rebuild

---

---

**Algorithm 7** Rebuild

---

1: $i \leftarrow 1$
2: **while** $i \leq t$ **and** $n_i^* < \tau n_i$ **do**
3:   $i \leftarrow i + 1$
4: **end while**
5: **if** $i \leq t$ **then**
6:   ConstructFromLayer($i$)
7: **end if**

---

**Lemma 3.3.** *The assignment $\sigma$ maintained by our dynamic algorithm always satisfies* $\mathrm{cost}(\sigma) = O(1) \cdot \mathrm{opt}(U)$.

**Answering a query:** Upon receiving a query, we consider a weighted $k$-median instance $(S, d, w)$, where each point $x \in S$ receives a weight $w(x) := |\sigma^{-1}(x)|$. Next, we compute a $O(1)$-approximate solution $S^* \subseteq S$, with $|S^*| \leq k$, to this weighted instance, so that $\mathrm{cost}_w(S^*, S) \leq O(1) \cdot \mathrm{opt}_w(S)$. We then return the centers in $S^*$.

**Corollary 3.4.** $\mathrm{cost}(S^*) = O(1) \cdot \mathrm{opt}(U)$.

Corollary 3.4 implies that our dynamic algorithm has $O(1)$ approximation ratio. It holds because of Lemma 3.3, and its proof immediately follows from [19]. We delegate this proof to Appendix C.

**Corollary 3.5.** *Our algorithm has $\tilde{O}(k^2)$ query time.*

*Proof* (Sketch). By Lemma 3.2, we have $t = \tilde{O}(1)$. Since the dynamic algorithm maintains at most $\tilde{O}(k)$ centers $S_i$ in each layer $i$ (follows from Invariant 3.1), we have $|S| = \sum_{i \in [t]} |S_i| = \tilde{O}(k)$. Using appropriate data structures (see the proof of Claim 3.7), in $O(1)$ time we can find the number of points assigned to any center $x \in S$ under $\sigma$ (given by $|\sigma^{-1}(x)| = w(x)$). Thus, the weighted $k$-median instance $(S, d, w)$ is of size $\tilde{O}(k)$, and upon receiving a query we can construct the instance in $\tilde{O}(k)$ time. We now run a static $O(1)$-approximation algorithm [27] on $(S, d, w)$, which returns the set $S^*$ in $\tilde{O}(k^2)$ time. $\qquad\square$

**Lemma 3.6.** *Our algorithm has $\tilde{O}(k)$ update time.*

Corollaries 3.4, 3.5 and Lemma 3.6 imply Theorem 1.1. It now remains to prove Lemma 3.6.

### 3.3 Proof of Lemma 3.6

We first bound the time taken to handle an update, excluding the time spent on rebuilding the layers.

**Claim 3.7.** *Excluding the calls to* `Rebuild`*, Algorithm 5 and Algorithm 6 both run in* $\tilde{O}(1)$ *time.*

*Proof.* We first describe how our dynamic algorithm maintains the assignment $\sigma$ implicitly. In each layer $i \in [t]$, the assignment $\sigma_i$ partitions the set $C_i$ into $|S_i|$ *clusters* $\{\sigma_i^{-1}(x)\}_{x \in S_i}$. For each such cluster $Z = \sigma_i^{-1}(x)$, we maintain: (1) a unique id, given by $\mathrm{id}(Z)$, (2) its center, given by $\mathrm{center}(\mathrm{id}(Z)) := x$, and (3) its size, given by $\mathrm{size}(\mathrm{id}(Z)) := |Z|$. For each point $y \in Z$, we also maintain the id of the cluster it belongs to, given by $\mathrm{cluster}(y) := \mathrm{id}(Z)$. Using these data structures, for any $y \in C_i$ we can report in $O(1)$ time the center $\sigma_i(y)$, and we can also report the size of a cluster in $O(1)$ time.

Recall that $t = \tilde{O}(1)$ as per Lemma 3.2. Thus, Algorithm 5 takes $\tilde{O}(1)$ time, excluding the call to `Rebuild` in Line 6. For Algorithm 6, the key thing to note is that using the data structures described above, Lines 8 - 10 can be implemented in $O(1)$ time, by setting $y$ as the new center of the cluster $\sigma_i^{-1}(x)$ and by decreasing the size of the cluster by one. It follows that excluding the call to `Rebuild` in Line 16, Algorithm 6 also runs in $\tilde{O}(1)$ time. $\square$

**Claim 3.8.** *A call to* `ConstructFromLayer`$(i)$*, as described in Algorithm 4, takes* $\tilde{O}(k \cdot |U_i|)$ *time.*

*Proof* (Sketch). By Lemma 3.2, the **while** loop in Algorithm 4 runs for $\tilde{O}(1)$ iterations. Hence, within a $\tilde{O}(1)$ factor, the runtime of Algorithm 4 is dominated by the call to `AlmostCover`$(U_j)$ in Line 4. Accordingly, we now consider Algorithm 2. As the ball $B_{U'}(X, r)$ can be computed in $O(|X| \cdot |U'|)$ time, using a binary search the value $\nu'$ (see Line 2) can be found in $\tilde{O}(k \cdot |U'|)$ time. The rest of the steps in Algorithm 2 also take $\tilde{O}(k \cdot |U'|)$ time. Thus, it takes $\tilde{O}(k \cdot |U_j|)$ time to implement Line 4 of Algorithm 4. Hence, the total runtime of Algorithm 4 is $\sum_{j \in [i,t]} \tilde{O}(k \cdot |U_j|) = \sum_{j \in [i,t]} \tilde{O}(k \cdot |U_i|) = \tilde{O}(k \cdot |U_i|)$. $\square$

Define the potential $\Phi := \sum_{i \in [t]} n_i^*$. For each update in $U$, the potential $\Phi$ increases by at most $t$, excluding the calls to `Rebuild` (see Algorithm 5 and Algorithm 6). Thus, by Lemma 3.2, each update increases the value of $\Phi$ by at most $\tilde{O}(1)$. Now, consider any call to `ReconstructFromLayer`$(i)$. Algorithm 7 and Invariant 3.1 ensure that just before this call, we had $n_i^* \geq \tau n_i = \Omega(|U_i|)$, since $\tau$ is a constant. Hence, because of Line 3 in Algorithm 4, during this call the value of $\Phi$ decreases by at least $\Omega(|U_i|)$. Claim 3.8, in contrast, implies that the time taken to implement this call is $\tilde{O}(k \cdot |U_i|)$.

To summarize, each update in $U$ creates $\tilde{O}(1)$ units of potential, and the total time spent on the calls to `Rebuild` is at most $\tilde{O}(k)$ times the decrease in the potential. Since the potential $\Phi$ is always nonnegative, it follows that the amortized time spent on the calls to `Rebuild` is $\tilde{O}(k)$. This observation, along with Claim 3.7, implies Lemma 3.6.

## 4   Experimental Evaluation

**Datasets.** We conduct the empirical evaluation of our algorithm on five datasets from the UCI repository [17]: (1) Census [23] with $2,458,285$ points of dimension 68, (2) KDD-Cup [32] containing $311,029$ points of dimension 74, (3) Song [5] with $515,345$ points of dimension 90, (4) Drift [33, 30] with $13,910$ points of dimension 129, and (5) SIFT10M [17] with $11,164,866$ points of dimension 128.[3]

We generate the dynamic instances that we use in our study as follows. We keep the first $10,000$ points of each dataset, in the order that they appear in their file. This choice allows us to test against competing algorithms that are not as efficient, and at the same time captures the different practical aspects of the algorithms that we consider; as a sanity check, we perform an experiment on an instance that is an order of magnitude larger to confirm the scalability of our algorithm, and that the relative behavior in terms of the cost of the solutions remains the same among the tested algorithms.

---

[3]We are not aware of any dynamic dataset containing the sequence of arrivals and that is publicly available. We generated sequences of updates following a typical setting used in studies of dynamic algorithms (e.g., [16]).

We use the $L_2$ distance of the embeddings of two points to compute their distance. To avoid zero distance without altering the structural properties of the instances, we add $1/|U|$ to all distances, where $U$ is the set of points of the instance.

**Order of Updates.** We obtain a dynamic sequence of updates following the sliding window approach. In this typical approach, one defines a parameter $\kappa$ indicating the size of the window, and then "slides" the window over the static sequence of points in steps creating at each step $i$ a point insertion of the $i$-th point (if there are more than $i$ points) and a point deletion of the $(i - \kappa)$-th point (if $\kappa > i$), until each point of the static instance is inserted once and deleted once. We set $\kappa = 2,000$, which results in our dynamic instances having at most $2,000$ points at any point in time.

**Algorithms.** We compare our algorithm from Section 3 against the state-of-the-art algorithm for maintaining a solution to the $k$-median problem, which is obtained by dynamically maintaining a coreset for the same problem [20]. In particular, [20] presented an algorithm that can dynamically maintain an $\epsilon$-coreset of size $O(\epsilon^{-2}k \operatorname{poly}(\log n, \log(1/\epsilon)))$ for the $k$-median and the $k$-means problems. Their algorithm has a worst-case update time of $O(\epsilon^{-2}k^2 \operatorname{poly}(\log n, \log(1/\epsilon)))$. To the best of our knowledge, this is the first implementation of the algorithm in [20]. For brevity, we refer to our algorithm as OURALG, and the algorithm from [20] as HK.

Since the exact constants and thresholds required to adhere to the theoretical guarantees of [20] are impractically large, there is no obvious way to use the thresholds provided by the algorithm in practice without making significant modifications. In order to give a fair comparison of the two algorithms, we implemented each of them to have a single parameter controlling a trade-off between update time, query time, and the cost of the solution. OURALG has a parameter $\phi$ which we set to be the number of points sampled during the construction of a layer in Algorithm 2. In other words, we sample $\phi$ points in Line 1 of Algorithm 2 instead of $\alpha k'$. We fix the constants $\epsilon$ and $\beta$ used by our algorithm (see Section 3) to $0.2$ and $0.5$ respectively. HK has a parameter $\psi$ which we set to be the number of points sampled during the static coreset construction from [9] which is used as a black box, replacing the large threshold required to adhere to the theoretical guarantees. Since this threshold is replaced by a single parameter, it also replaces the other parameters used by HK, which are only used in this thresholding process. For the bicriteria algorithm required by this static coreset construction, we give the same implementation as the empirical evaluations in [9] and use `kmeans++` with 2 iterations of Lloyd's.

For each of $\phi$ and $\psi$, we experimented with the values $250, 500, 1000$. As the values of $\phi$ and $\psi$ are increased, the asymptotic update times and query times of the corresponding algorithms increase, while the cost of the solution decreases.

**Setup.** All of our code is written in Java and is available online.[4] We did not use parallelization. We used a machine with 8 cores, a 2.9Ghz processor, and 16 GiB of main memory.

**Results.** We compare OURALG($\phi = 500$) against HK($\psi = 1000$). We selected these parameters such that they obtain a good solution quality, without these parameters being unreasonably high w.r.t. the size of our datasets. Our experiments suggest that the solution quality produced by OURALG is robust against different values of $\phi$ (typically these versions of the algorithm differ by less than $1\%$), while that of HK is more sensitive (in several instances the differences are in the range of $3 - 9\%$, while for KDD-Cup HK($\psi = 250$) produces much worse solutions). The update time among the different versions of each algorithm do not differ significantly. Lastly, the query time of OURALG is less sensitive compared to the query time of HK. Specifically, the average query time of OURALG($\phi = 1000$) is roughly 3 times slower than OURALG($\phi = 250$), while the average query time of HK($\psi = 1000$) is 11 times slower compared to HK($\psi = 250$). For OURALG we selected the middle value $\phi = 500$. Since the solution quality of HK drops significantly when using smaller values of $\psi$, we selected $\psi = 1000$ to prioritize for the quality of the solution produced by the algorithm. Since we did not observe significant difference in the behavior across the datasets, we tuned these parameters considering three or our datasets. We provide the relevant plots in Appendix E.4.

**Update Time Evaluation.** In Figure 1 (left) we plot the total update time over the sequence of updates for the dataset Song, and only for $k = 50$. OURALG runs up to more than 2 orders of

---

[4]`https://github.com/martin-costa/NeurIPS23-dynamic-k-clustering`

magnitude faster compared to HK, independently of their parameter setup (see Appendix E for more details). The relative performance of the algorithms is similar for all datasets and choices of $k$; we summarize the results in Table 1, while the corresponding plots can be found in Appendix E.

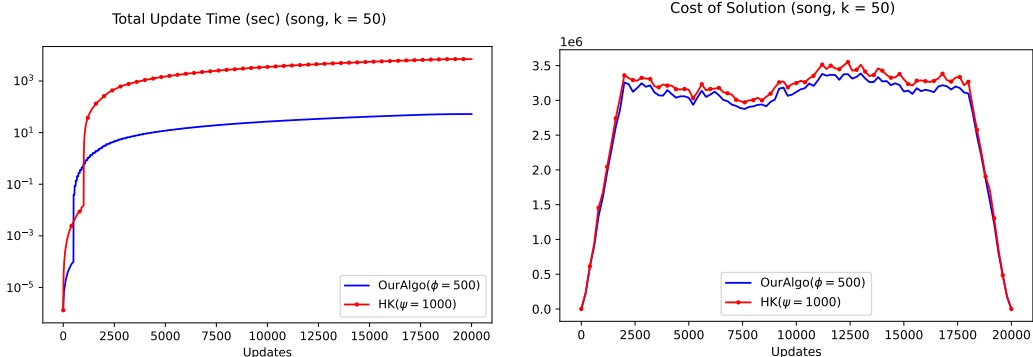

Figure 1: The total update time in seconds (left) and the solution cost (right) for OURALG($\phi = 500$) and HK($\psi = 1000$) on Song, for $k = 50$.

Table 1: The ratio of the total update time incurred by HK($\psi = 1000$) divided by the time for OURALG($\phi = 500$), for the five datasets that we considered and for $k \in \{10, 50, 100\}$.

|         | Song    | Census  | KDD-Cup | Drift   | SIFT10M |
|---------|---------|---------|---------|---------|---------|
| $k = 10$  | 31.774  | 32.172  | 62.776  | 32.408  | 32.857  |
| $k = 50$  | 134.850 | 131.333 | 201.147 | 140.826 | 142.989 |
| $k = 100$ | 262.427 | 255.927 | 377.350 | 276.848 | 280.925 |

**Solution Cost Evaluation.** Figure 1 (right) shows the cost of the solution obtained by OURALG($\phi = 500$) and HK($\psi = 1000$), on dataset Song for $k = 50$. In general, the two algorithms have similar performance with OURALG($\phi = 500$) producing slightly better ($1.3\% - 6.5\%$) solutions on most instances; the exceptions being on the KDD-Cup dataset, where for $k = 10$ HK($\psi = 1000$) performs $1.2\%$ better, and for $k = 100$ where OURALG($\phi = 500$) performs $35\%$ better. We summarize their relevant performance on the different datasets in Table 2, and provide the complete set of plots in the Appendix.

Table 2: The mean ratio of the solution cost for HK($\psi = 1000$) divided by that of OURALG($\phi = 500$) averaged over the updates, for the five datasets that we considered and for $k \in \{10, 50, 100\}$.

|         | Song  | Census | KDD-Cup | Drift | SIFT10M |
|---------|-------|--------|---------|-------|---------|
| $k = 10$  | 1.013 | 1.018  | 0.988   | 1.020 | 1.014   |
| $k = 50$  | 1.036 | 1.056  | 1.019   | 1.037 | 1.032   |
| $k = 100$ | 1.049 | 1.065  | 1.538   | 1.059 | 1.038   |

**Query Time Evaluation.** Finally, we compare OURALG($\phi = 500$) to HK($\psi = 1000$) in terms of their query time. To measure query time, we evenly distribute 100 queries across the update sequence and measure the average query time. In Table 3 we report the average ratio of the query times obtained by OURALG($\phi = 500$) divided by that of HK($\psi = 1000$) over the queries executed on each of the datasets and parameters of $k$ that we considered. The query time obtained by the two algorithms is typically within a factor 2 of each other. OURALG($\phi = 500$) performs better on KDD-Cup, while HK($\psi = 1000$) performs better on rest of the datasets. This inconsistency is due to differences in the structure of the datasets (we also tried randomizing the order of the updates, and it didn't change the relative picture w.r.t. query time).

**Longer Sequence of Updates.** While the above experiments capture well the different aspects of the two algorithms, we further conducted an experiment on the KDD-Cup dataset where we consider

Table 3: The mean ratio of query times for $\mathrm{HK}(\psi = 1000)$ divided by that of $\mathrm{OURALG}(\phi = 500)$, for the five datasets that we considered and for $k \in \{10, 50, 100\}$.

|           | Song  | Census | KDD-Cup | Drift | SIFT10M |
|-----------|-------|--------|---------|-------|---------|
| $k = 10$  | 0.575 | 0.586  | 2.629   | 0.577 | 0.572   |
| $k = 50$  | 0.568 | 0.577  | 1.888   | 0.570 | 0.564   |
| $k = 100$ | 0.560 | 0.573  | 1.543   | 0.564 | 0.558   |

a window size of $\kappa = 5,000$ and applied it to the first $100,000$ points. As expected the relative performance in terms of solution cost and query time remains the same, while the gap in terms of update time grows significantly ($\mathrm{HK}(\psi = 1000)$ performs more than 3 orders of magnitude worse than $\mathrm{OURALG}(\phi = 500)$). The corresponding plots and numbers are provided in the Appendix.

**Conclusion of the Experiments.** Our experimental evaluation shows the practicality of our algorithms on a set of five standard datasets used for the evaluation of dynamic algorithms in metric spaces. In particular, our experiments verify the theoretically superior update time of our algorithm w.r.t. to the state-of-the-art algorithm for the fully-dynamic $k$-median problem [20], where our algorithm performs orders of magnitude faster than HK. On top of that, the solution quality produced by our algorithm is better than HK in most settings, and the query times of the two algorithms are comparable. We believe that our algorithm is the best choice when it comes to efficiently maintaining a dynamic solution for the $k$-median problem in practice.

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

# A Extension to $k$-Means and $(k, p)$-Clustering

As stated in [21, 8], while [28] only discusses the $k$-median problem, their construction can easily be modified to work for $k$-means clustering and further generalized to work for $(k, p)$-clustering, where the $(k, p)$-clustering problem is defined in the same way as $k$-median problem except that we want to minimize $\sum_{x \in U} d(x, S)^p$ for some $S \subseteq U$ of size at most $k$. Note that $(k, 1)$-clustering and $(k, 2)$-clustering correspond to $k$-median and $k$-means respectively.

We define a $\rho$-metric space $(U, d)$ in the same way as a metric space except for relaxing the condition that $d$ must satisfy the triangle inequality to the condition that $d(x, y) \leq \rho(d(x, z) + d(z, y))$ for all $x, y, z \in U$. Given a metric space $(U, d)$ and some $p \geq 1$, the results in Section 6 of [9] can easily be used to show that $(U, d^p)$ is a $2^{p-1}$-metric space, where $d^p(x, y)$ is defined to be $d(x, y)^p$.

We now show that the assignment $\sigma$ maintained by our algorithm is $O(\rho^3)$-approximate when $U$ is a $\rho$-metric space (i.e. that $\texttt{cost}(\sigma) = O(\rho^3) \cdot \texttt{opt}(U)$) and that the extraction technique of [19] can be generalized to $\rho$-metric spaces.

**Lemma A.1.** *When the underlying space $U$ is a $\rho$-metric space, the assignment $\sigma$ maintained by our algorithm is $O(\rho^3)$-approximate.*

*Proof.* By making the appropriate changes to the proofs of Lemma B.3 and Lemma B.4, we get generalizations of these lemmas to $\rho$-metric spaces, where the lemma statements are the same except for an extra $\rho$ factor in the inequalities.

**Lemma A.2.** *Given any positive $\xi$, there exists a sufficiently large choice of $\alpha$ such that $\nu_i \leq 2\rho \cdot \mu_\gamma(U_i^{\text{OLD}})$ for each $i \in [t-1]$ with probability at least $1 - e^{-\xi k'}$.*

**Lemma A.3.** *Given metric subspaces $U_1$ and $U_2$ of $U$ such that $|U_1 \oplus U_2| \leq \epsilon\gamma|U_1|$, we have that $\mu_\gamma(U_1) \leq 2\rho \cdot \mu_{\gamma^*}(U_2)$.*

These two lemmas immediately imply the following generalization of Lemma B.5.

**Lemma A.4.** $\nu_i \leq 4\rho^2 \cdot \mu_{\gamma^*}(U_i)$ *for each $i \in [t-1]$ with probability at least $1 - e^{-\xi k'}$.*

The upper bound on $\texttt{cost}(\sigma)$ given in Lemma B.6 can be generalized by noticing that $\texttt{cost}(\sigma, C_i) \leq 2\rho\nu_i|C_i|$ for all $i \in [t-1]$, which us that

$$\texttt{cost}(\sigma) \leq \sum_{i=1}^{t} 2\rho\nu_i|C_i|.$$

The lower bound on $\texttt{opt}(U)$ given in Lemma B.10 holds for $\rho$-metric spaces with no modifications. Hence, we get that with probability at least $1 - e^{-\xi k'}$ we have that

$$\texttt{cost}(\sigma) \leq \sum_{i=1}^{t} 2\rho\nu_i|C_i| \leq \sum_{i=1}^{t} 8\rho^3\mu_i|C_i| \leq \frac{16\rho^3 r}{1 - \gamma^*} \texttt{cost}(S).$$

$\square$

By making the appropriate modifications to the proof of Theorem C.1, we can extend this theorem to work for $\rho$-metric spaces. In particular, we can obtain a proof of Theorem A.5 by taking the proof of Theorem C.1 and adding extra $\rho$ factors whenever the triangle inequality is applied.

**Theorem A.5.** *Given a $\phi$-approximate $m$-assignment $\pi : U \to U$, any $\psi$-approximate solution to the weighted $k$-median instance $(\pi(U), d, w)$, where each point $x \in \pi(U)$ receives weight $w(x) := |\pi^{-1}(x)|$, is also a $\left(\phi\rho + 2(1+\phi)\psi\rho^3\right)$-approximate solution to the $k$-median instance $(U, d)$ where $U$ is a $\rho$-metric space.*

Since the algorithm in [27] is $O(1)$-approximate on $O(1)$-metric spaces, it immediately follows by applying Theorem A.5 and Lemma A.1 that our algorithm maintains a $O(1)$-approximate solution to the $k$-median problem on $(U, d^p)$ for $p = O(1)$. Since the $k$-median problem on $(U, d^p)$ is exactly the $(k, p)$-clustering problem on $(U, d)$, it follows that our algorithm generalizes to solve instances of $(k, p)$-clustering in metric spaces.

# B  Proofs of Lemma 3.2 and Lemma 3.3

Throughout this section, we fix $\gamma$ to be any real such that $\beta < \gamma < 1$ and $\epsilon$ to be any real such that $0 < \epsilon < \min\{\frac{1-\gamma}{2\gamma}, 1\}$. Let $\beta^*$ and $\gamma^*$ denote $\beta(1 - \epsilon)$ and $\gamma(1 + 2\epsilon)$ respectively.

## B.1  Proof of Lemma 3.2

We first prove Lemma B.1, which shows that the sizes of the sets $U_i$ decrease exponentially with $i$.

**Lemma B.1.** *For all $i \in [t - 1]$, $|U_{i+1}| \leq (1 - \beta^*)|U_i|$.*

*Proof.* Consider the ratio $|U_{i+1}|/|U_i|$. Since $U_{i+1} \subseteq U_i$ and $U_{i+1}$ is reconstructed every time $U_i$ is reconstructed, it follows that $|U_{i+1}|/|U_i|$ is at most $(n_{i+1} + \ell)/(n_i + \ell - \ell')$, where $n_j$ is the size of $U_j$ at the time it was last reconstructed and $\ell$ and $\ell'$ are the number of insertions and deletions that have occurred since the last time $U_{i+1}$ was reconstructed respectively. By Lemma B.2, we get that this expression is upper bounded by $(n_{i+1} + \tau n_{i+1})/n_i$. Now we can observe that

$$\frac{|U_{i+1}|}{|U_i|} \leq \frac{n_{i+1} + \ell}{n_i + \ell - \ell'} \leq \frac{n_{i+1} + \tau n_{i+1}}{n_i} \leq \frac{n_{i+1}}{n_i} + \tau \leq (1 - \beta) + \epsilon\beta = 1 - \beta^*,$$

where we use the facts that $n_{i+1} \leq (1 - \beta)n_i$ and $\tau \leq \epsilon\beta$ in the final inequality.

**Lemma B.2.** *Given some integer $i \in [t - 1]$, let $\ell$ and $\ell'$ be the number of insertions and deletions that have occurred since the last time $U_{i+1}$ was reconstructed respectively. Then we have that*

$$\frac{n_{i+1} + \ell}{n_i + \ell - \ell'} \leq \frac{n_{i+1} + \tau n_{i+1}}{n_i}.$$

*Proof.* First, note that $(n_{i+1} + \ell)/(n_i + \ell - \ell') \leq (n_{i+1} + \ell)/(n_i - \ell')$. Now, given some reals $A \geq a \geq 0$ and $0 \leq N \leq A - a$, we define a function $f : [0, 1] \rightarrow \mathbb{R}$ by $f(x) = (a + xN)/(A - (1 - x)N)$. The derivative of $f$ is $-N(a - A + N)/((x - 1)N + A)^2$ and is non-negative for all $x \in [0, 1]$. Hence, $f(x) \leq f(1)$ for all $x \in [0, 1]$.

By setting $A = n_i$, $a = n_{i+1}$, $N = \ell + \ell'$ and noting that $\ell + \ell' \leq \tau n_{i+1}$ by Invariant 3.1 and $n_{i+1} \leq (1 - \beta)n_i$, we get that

$$\ell + \ell' \leq \tau n_{i+1} \leq \beta n_{i+1} = (1 + \beta)n_{i+1} - n_{i+1} \leq (1 + \beta)(1 - \beta)n_i - n_{i+1} \leq n_i - n_{i+1},$$

and hence it follows that

$$\frac{n_{i+1} + \ell}{n_i - \ell'} = f\left(\frac{\ell}{\ell + \ell'}\right) \leq f(1) = \frac{n_{i+1} + \ell + \ell'}{n_i} \leq \frac{n_{i+1} + \tau n_{i+1}}{n_i}.$$

$\square$

$\square$

Since $|U_1| = |U|$, $|U_{t-1}| > (1 - \tau)\alpha k' = \Omega(k)$, and $\beta^*$ is a constant, it follows from Lemma B.1 that $t = O\left(\log(|U|/k)\right) = \tilde{O}(1)$.

## B.2  Proof of Lemma 3.3

**Bounding the Radii $\nu_i$ (Lemma B.5).**  Let $U_i^{\mathrm{OLD}}$ denote the state of the $i$th layer the last time it was reconstructed for $i \in [t]$. We now use the following crucial lemma which is analogous to Lemma 4.3.3 in [26].

**Lemma B.3.** *Given any positive $\xi$, there exists a sufficiently large choice of $\alpha$ such that $\nu_i \leq 2\mu_\gamma(U_i^{\mathrm{OLD}})$ for each $i \in [t - 1]$ with probability at least $1 - e^{-\xi k'}$.*

Henceforth, we fix some positive $\xi$ and sufficiently large $\alpha$ such that Lemma B.3 holds.

**Lemma B.4.** *Given metric subspaces $U_1$ and $U_2$ of $U$ such that $|U_1 \oplus U_2| \leq \epsilon\gamma|U_1|$, we have that $\mu_\gamma(U_1) \leq 2\mu_{\gamma^*}(U_2)$.[5]*

---

[5] $\oplus$ denotes symmetric difference, i.e. $U_1 \oplus U_2 = (U_1 \setminus U_2) \cup (U_2 \setminus U_1)$.

*Proof.* Let $X$ be a subset of $U_2$ of size $k$ such that $\nu_{\gamma(1+2\epsilon)}(X, U_2) = \mu_{\gamma(1+2\epsilon)}(U_2)$, $\rho = \mu_{\gamma(1+2\epsilon)}(U_2)$, and $A = B_{U_1}(X, \rho)$. Now note that

$$
\begin{aligned}
|A| &= |B_{U_1 \cup U_2}(X, \rho) \setminus B_{U_2 \setminus U_1}(X, \rho)| \\
&\geq |B_{U_2}(X, \rho)| - |B_{U_2 \setminus U_1}(X, \rho)| \\
&\geq \gamma(1 + 2\epsilon)|U_2| - |U_2 \setminus U_1| \\
&\geq \gamma(1 + 2\epsilon)|U_2| - \epsilon\gamma|U_1| \\
&\geq \gamma(1 + 2\epsilon)(|U_1| - \epsilon\gamma|U_1|) - \epsilon\gamma|U_1| \\
&= \gamma|U_1| + \epsilon\gamma(1 - \gamma(1 + 2\epsilon))|U_1| \\
&\geq \gamma|U_1|.
\end{aligned}
$$

Since there also exists a subset $Y \subseteq A$ of size $k$ such that $A \subseteq B_{U_1}(Y, 2\rho)$, it follows that $\nu_\gamma(Y, U_1) \leq 2\rho$. Hence, $\mu_\gamma(U_1) \leq \nu_\gamma(Y, U_1) \leq 2\mu_{\gamma(1+2\epsilon)}(U_2)$. □

**Lemma B.5.** $\nu_i \leq 4\mu_{\gamma^*}(U_i)$ *for each* $i \in [t-1]$ *with probability at least* $1 - e^{-\xi k'}$.

*Proof.* For each $i \in [t-1]$, $|U_i \oplus U_i^{\text{OLD}}| \leq \tau|U_i^{\text{OLD}}|$ since, by Invariant 3.1, at most $\tau|U_i^{\text{OLD}}|$ points have been inserted or deleted from $U_i$ since it was last reconstructed. Noticing that $\tau \leq \epsilon\gamma$, we can see that

$$
|U_i \oplus U_i^{\text{OLD}}| \leq \epsilon\gamma|U_i^{\text{OLD}}|.
$$

By now applying Lemma B.4 it follows that $\mu_\gamma(U_i^{\text{OLD}}) \leq 2\mu_{\gamma^*}(U_i)$. The lemma follows by combining this result with Lemma B.3. □

**Upper Bounding $\text{cost}(\sigma)$ (Lemma B.6).**

**Lemma B.6.**

$$
\text{cost}(\sigma) \leq \sum_{i=1}^{t} 2\nu_i|C_i|.
$$

*Proof.* We first note that for all $i \in [t-1]$, $\text{cost}(\sigma, C_i) \leq 2\nu_i|C_i|$. This follows directly from the fact that each point $x$ in $C_i$ is assigned to some point $y \in C_i$ such that $d(x, y) \leq 2\nu_i$. Since the $C_i$ partition $U$ and $\text{cost}(\sigma, C_t) = 0$, we get:

$$
\text{cost}(\sigma) = \sum_{i=1}^{t} \text{cost}(\sigma, C_i) \leq \sum_{i=1}^{t} 2\nu_i|C_i|.
$$

□

**Lower Bounding $\text{opt}(U)$ (Lemma B.10).** Let $r$ denote $\lceil \log_{1-\beta^*} \frac{1-\gamma^*}{3} \rceil$ and for each $i \in [t]$ let $\mu_i$ denote $\mu_{\gamma^*}(U_i)$.

For the rest of this subsection we fix an arbitrary $S \subseteq U$ of size $k$. For each $i \in [t]$, let $F_i$ denote the set $\{x \in U_i \mid d(x, S) \geq \mu_i\}$, and for any integer $m > 0$, let $F_i^m$ denote $F_i \setminus (\cup_{j>0} F_{i+jm})$ and $G_{i,m}$ denote the set of all integers $j \in [t]$ and $j \equiv i \pmod{m}$.

**Lemma B.7.** *Given some $i \in [t]$ and a subset $X \subseteq F_i$, we have that $|F_i| \geq (1 - \gamma^*)|U_i|$ and $cost(S, X) \geq \mu_i|X|$.*

*Proof.* It follows directly from the definition of $\mu_i$ that we have that $|F_i| \geq (1 - \gamma^*)|U_i|$. By the definition of $F_i$, we have that $\text{cost}(S, X) = \sum_{x \in X} d(x, S) \geq \mu_i|X|$. □

The following lemma is proven in [26].

**Lemma B.8** ([26], Lemma 4.3.8). *Given integers $\ell \in [t]$ and $m > 0$, we have that*

$$
\text{cost}(S, \cup_{i \in G_{\ell,m}} F_i^m) \geq \sum_{i \in G_{\ell,m}} \mu_i|F_i^m|.
$$

**Lemma B.9.** *For all $i \in [t-1]$, we have that $|F_i^r| \geq \frac{1}{2}|F_i|$.*

*Proof.* We first note that for all $i \in [t-r]$, we have that $|F_{i+r}| \leq \frac{1}{3}|F_i|$. This follows from the fact that

$$|F_{i+r}| \leq |U_{i+r}| \leq (1-\beta^*)^r|U_i| \leq \frac{(1-\beta^*)^r}{1-\gamma^*}|F_i| \leq \frac{1}{3}|F_i|,$$

where the first inequality follows from the fact that $F_{i+r} \subseteq U_{i+r}$, the second inequality follows from Lemma B.1, the third inequality follows from Lemma B.7, and the fourth inequality follows from the definition of $r$. We now get that

$$|F_i^r| = |F_i \setminus \cup_{j>0} F_{i+jr}| \geq |F_i| - \sum_{j>0} \frac{1}{3^j}|F_i| \geq \frac{1}{2}|F_i|.$$

$\square$

**Lemma B.10.**

$$\mathtt{cost}(S) \geq \frac{1-\gamma^*}{2r} \sum_{i=1}^{t} \mu_i |C_i|.$$

*Proof.* Let $\ell = \arg\max_{0 \leq \ell < r}\{\sum_{i \in G_{\ell,r}} \mu_i |F_i^r|\}$. Then we have that

$$\mathtt{cost}(S) \geq \mathtt{cost}(S, \cup_{i \in G_{\ell,r}} F_i^r) \geq \sum_{i \in G_{\ell,r}} \mu_i |F_i^r| \geq \frac{1}{r} \sum_{i=1}^{t} \mu_i |F_i^r| \geq \frac{1}{2r} \sum_{i=1}^{t} \mu_i |F_i|$$

$$\geq \frac{1-\gamma^*}{2r} \sum_{i=1}^{t} \mu_i |U_i| \geq \frac{1-\gamma^*}{2r} \sum_{i=1}^{t} \mu_i |C_i|.$$

The second inequality follows from Lemma B.8, the third inequality from averaging and the choice of $\ell$, the fourth inequality from Lemma B.9, and the fifth inequality from Lemma B.7. $\square$

**Proof of Lemma 3.3.** It follows that with probability at least $1 - e^{-\xi k'}$ we have that

$$\mathtt{cost}(\sigma) \leq \sum_{i=1}^{t} 2\nu_i |C_i| \leq \sum_{i=1}^{t} 8\mu_i |C_i| \leq \frac{16r}{1-\gamma^*} \mathtt{cost}(S)$$

for any set $S \subseteq U$ of size $k$. Hence, we have that

$$\mathtt{cost}(\sigma) \leq \frac{16r}{1-\gamma^*} \mathtt{opt}(U).$$

## C   Proof of Corollary 3.4

In order to prove this corollary, we apply the extraction technique presented in [28] (with full details appearing in [26]) which is a slight generalization of the techniques from [19]. In particular, we use the following theorem which follows as an immediate corollary of Theorem 6 in [26]. For completeness, we provide a proof of this theorem.

**Theorem C.1.** *Given a $\phi$-approximate $m$-assignment $\pi : U \to U$, any $\psi$-approximate solution to the weighted $k$-median instance $(\pi(U), d, w)$, where each point $x \in \pi(U)$ receives weight $w(x) := |\pi^{-1}(x)|$, is also a $(\phi + 2(1+\phi)\psi)$-approximate solution to the $k$-median instance $(U, d)$.*

*Proof.* Let $S^*$ be a solution to the weighted $k$-median instance $(\pi(U), d, w)$ and let $S$ be an optimal solution to the $k$-median instance $(U, d)$. Let $\phi$ and $\psi$ be constants such that $\mathtt{cost}(\pi, U) \leq \phi \cdot \mathtt{opt}(U)$ and $\mathtt{cost}_w(S^*, \pi(U)) \leq \psi \cdot \mathtt{opt}_w(\pi(U))$. We now show that $\mathtt{cost}(S^*, U) = O(1) \cdot \mathtt{opt}(U)$. We first note that

$$\mathtt{cost}(S^*, U) = \sum_{x \in U} d(x, S^*)$$

$$\leq \sum_{x \in U} d(x, \pi(x)) + \sum_{y \in \pi(U)} w(y) \cdot d(y, S^*)$$

$$= \mathtt{cost}(\pi, U) + \mathtt{cost}_w(S^*, \pi(U))$$

$$\leq \phi \cdot \mathtt{opt}(U) + \mathtt{cost}_w(S^*, \pi(U)).$$

Now note that, for any $X \subseteq U$ of size at most $k$, there exists some $Y \subseteq \pi(U)$ of size at most $k$ such that $\text{cost}_w(Y, \pi(U)) \leq 2 \cdot \text{cost}_w(X, \pi(U))$. Since $\text{cost}_w(S^*, \pi(U)) \leq \psi \cdot \text{cost}_w(Y, \pi(U))$ for all $Y \subseteq \pi(Y)$ of size at most $k$, we get the following.

$$\text{cost}_w(S^*, \pi(U)) \leq 2\psi \cdot \text{cost}_w(S, \pi(U))$$
$$= 2\psi \cdot \sum_{y \in \pi(U)} w(y) \cdot d(y, S)$$
$$= 2\psi \cdot \sum_{x \in U} d(\pi(x), S)$$
$$\leq 2\psi \cdot \sum_{x \in U} d(x, \pi(x)) + 2\psi \cdot \sum_{x \in U} d(x, S)$$
$$= 2\psi \cdot \text{cost}(\pi, U) + 2\psi \cdot \text{opt}(U)$$
$$\leq 2(1 + \phi)\psi \cdot \text{opt}(U).$$

By combining these two chains of inequalities, we get that

$$\text{cost}(S^*, U) \leq \phi \cdot \text{opt}(U) + \text{cost}_w(S^*, \pi(U)) \leq (\phi + 2(1 + \phi)\psi) \cdot \text{opt}(U).$$

$\square$

It immediately follows that we can get a $O(1)$-approximate solution to the instance $(U, d)$ by running a static weighted $k$-median algorithm on the instance $(\sigma(U), d, w)$.

## D   Lower Bounds on Update and Query Time

In the static (i.e. non-dynamic) setting, the $k$-median problem is defined as follows: given a metric space $U$, return a set $S$ of at most $k$ points from $U$ which minimizes the value of $\sum_{x \in S} d(x, S)$. The following lower bound for the static $k$-median problem is proven by Mettu in [26].

**Theorem D.1.** *Any $O(1)$-approximate randomized (static) algorithm for the $k$-median problem, which succeeds with even negligible probability, runs in time $\Omega(nk)$.*

Informally, the proof of this lower bound is obtained by constructing, for each $\delta > 0$, an input distribution of metric spaces (with polynomially bounded aspect ratio) on which no deterministic algorithm for the $k$-median problem succeeds with probability more than $\delta$. Theorem D.1 then follows by an application of Yao's minmax principle.

We can use this lower bound from the static setting in order to get a lower bound for the dynamic setting. First note that any incremental algorithm for $k$-median with amortized update time $u(n, k)$ and query time $q(n, k)$ can be used to construct a static algorithm for the $k$-median problem with running time $n \cdot u(n, k) + q(n, k)$ by inserting each point in the input metric space $U$ followed by a solution query. Hence, by Theorem D.1, we must have that $n \cdot u(n, k) + q(n, k) = \Omega(nk)$. Now assume that some incremental algorithm for $k$-median has query time $\tilde{O}(\text{poly}(k))$. If this algorithm also has an amortized update time of $\tilde{o}(k)$, then for the range of values of $k$ where $q(n, k) = \tilde{o}(nk)$, it follows that $\tilde{o}(nk)$ is $\Omega(nk)$, giving a contradiction. Hence, the amortized update time must be $\tilde{\Omega}(k)$ and Theorem D.2 follows.

**Theorem D.2.** *Any $O(1)$-approximate incremental algorithm for the $k$-median problem with $\tilde{O}(\text{poly}(k))$ query time must have $\tilde{\Omega}(k)$ amortized update time.*

It follows that the update time of our algorithm is optimal up to polylogarithmic factors.

# E   Omitted Experimental Results

## E.1   Update Time Evaluation

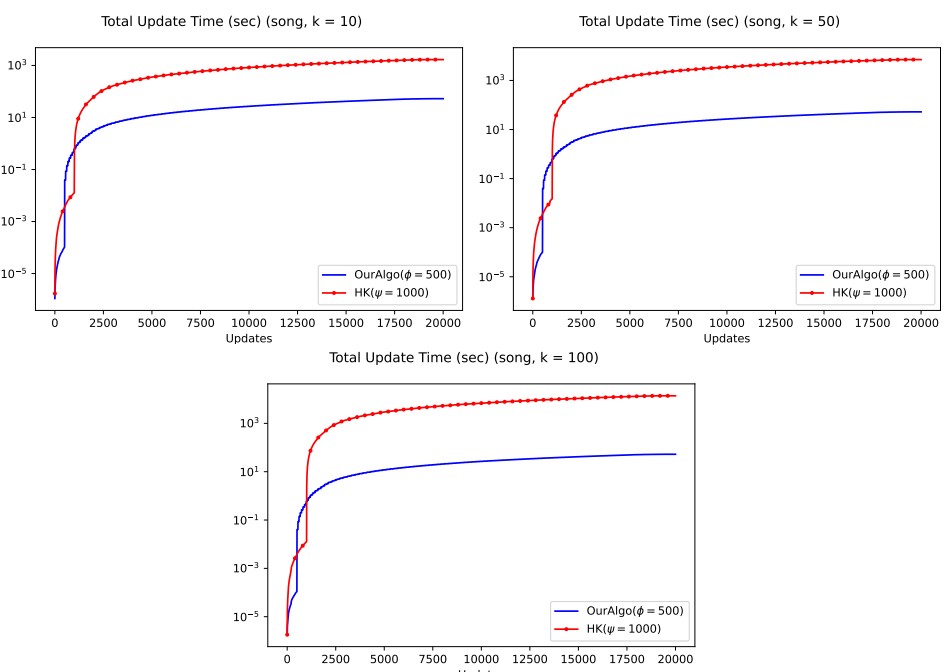

Figure 2: The cumulative update time for the different algorithms, on the $\mathsf{Song}$ dataset for $k = 10$ (top left), $k = 50$ (top right), $k = 100$ (bottom).

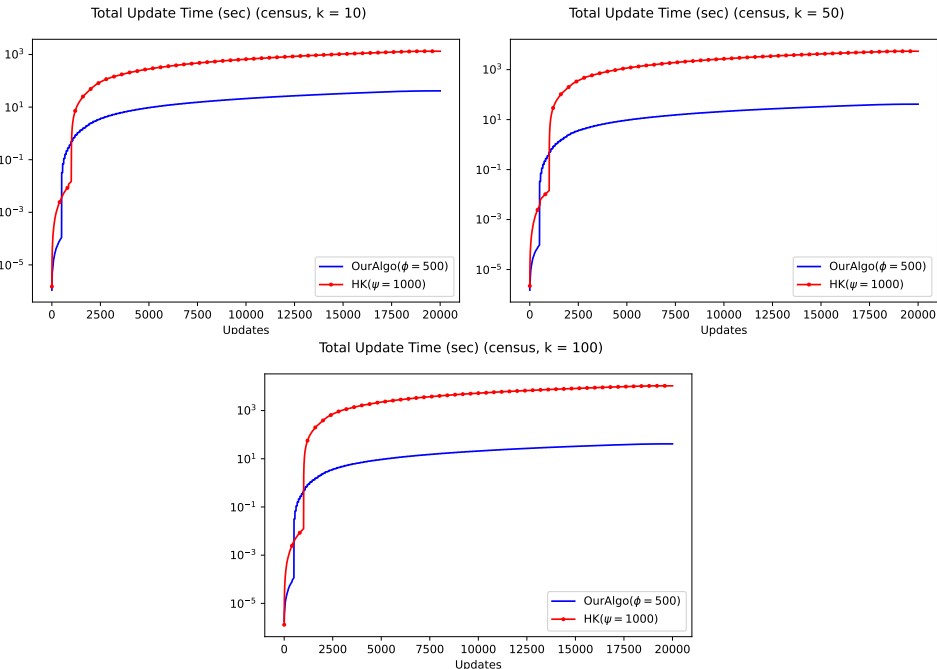

Figure 3: The cumulative update time for the different algorithms, on the $\mathsf{Census}$ dataset for $k = 10$ (top left), $k = 50$ (top right), $k = 100$ (bottom).

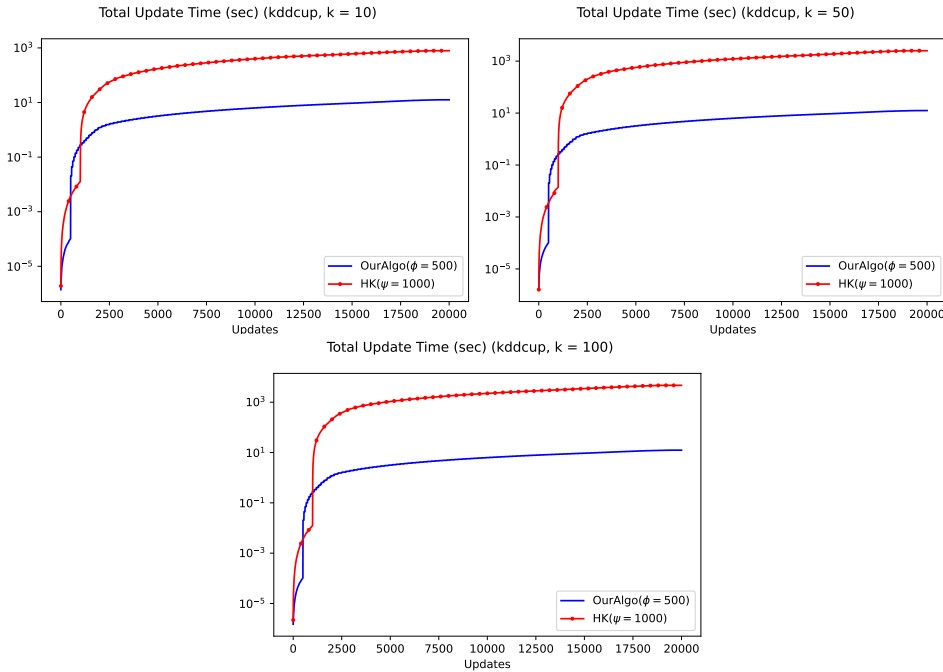

Figure 4: The cumulative update time for the different algorithms, on the KDD-Cup dataset for $k = 10$ (top left), $k = 50$ (top right), $k = 100$ (bottom).

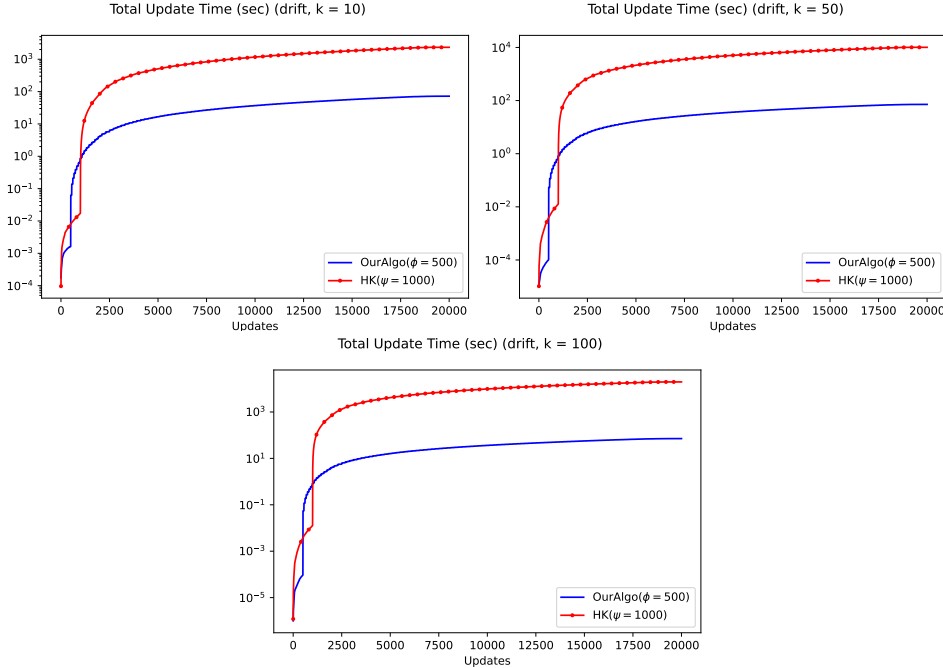

Figure 5: The cumulative update time for the different algorithms, on the Drift dataset for $k = 10$ (top left), $k = 50$ (top right), $k = 100$ (bottom).

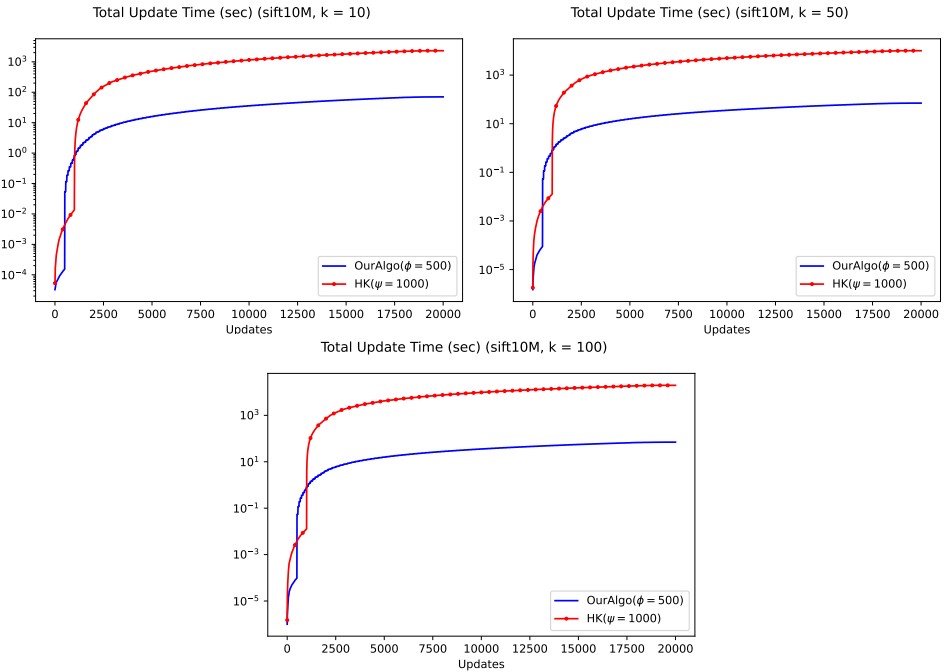

Figure 6: The cumulative update time for the different algorithms, on the SIFT10M dataset for $k = 10$ (top left), $k = 50$ (top right), $k = 100$ (bottom).

## E.2 Solution Cost Evaluation

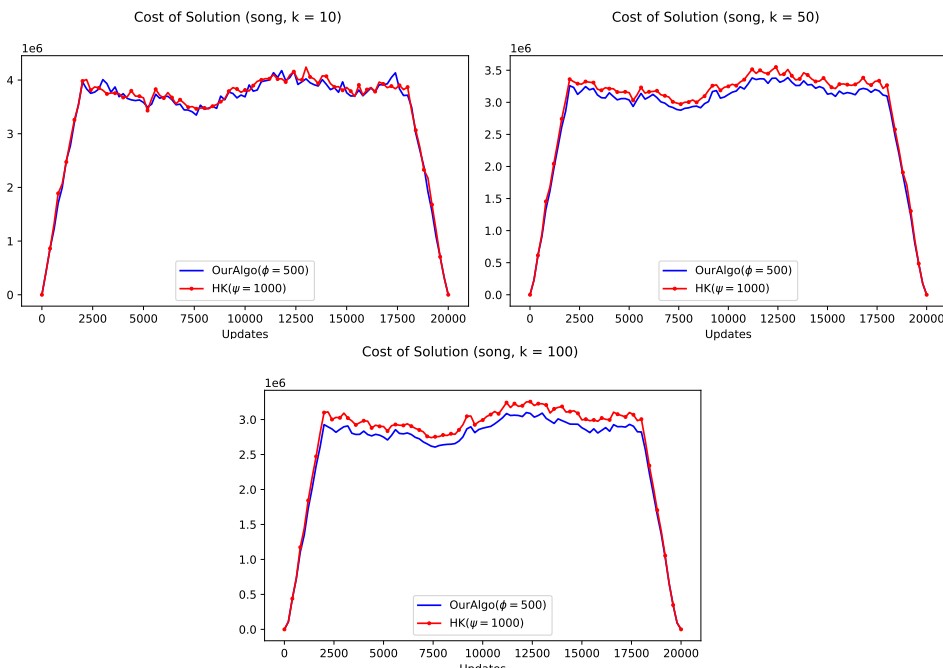

Figure 7: The solution cost by the different algorithms, on Song for $k = 10$ (top left), $k = 50$ (top right), $k = 100$ (bottom).

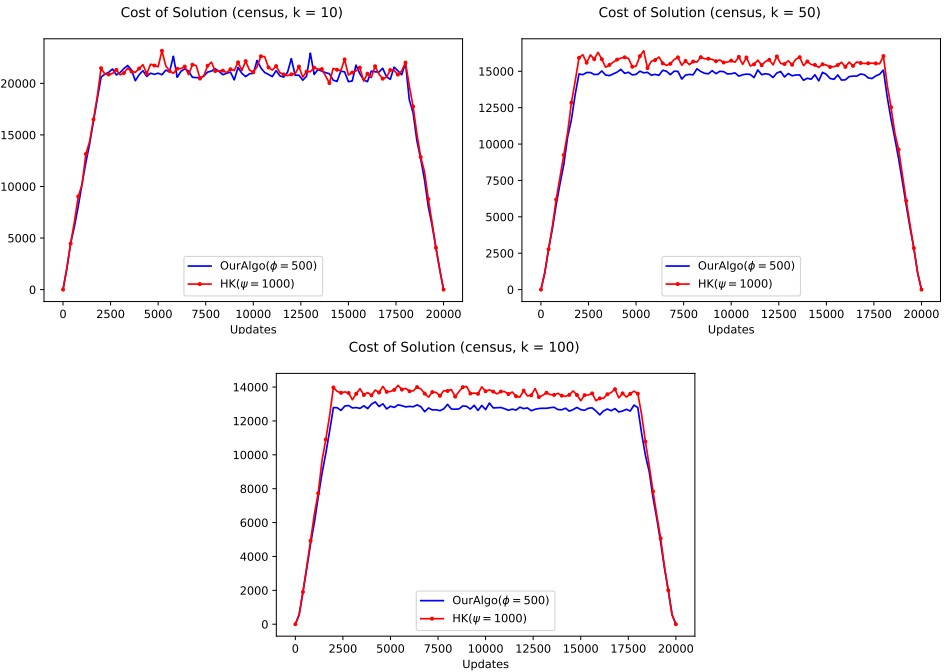

Figure 8: The solution cost by the different algorithms, on Census for $k = 10$ (top left), $k = 50$ (top right), $k = 100$ (bottom).

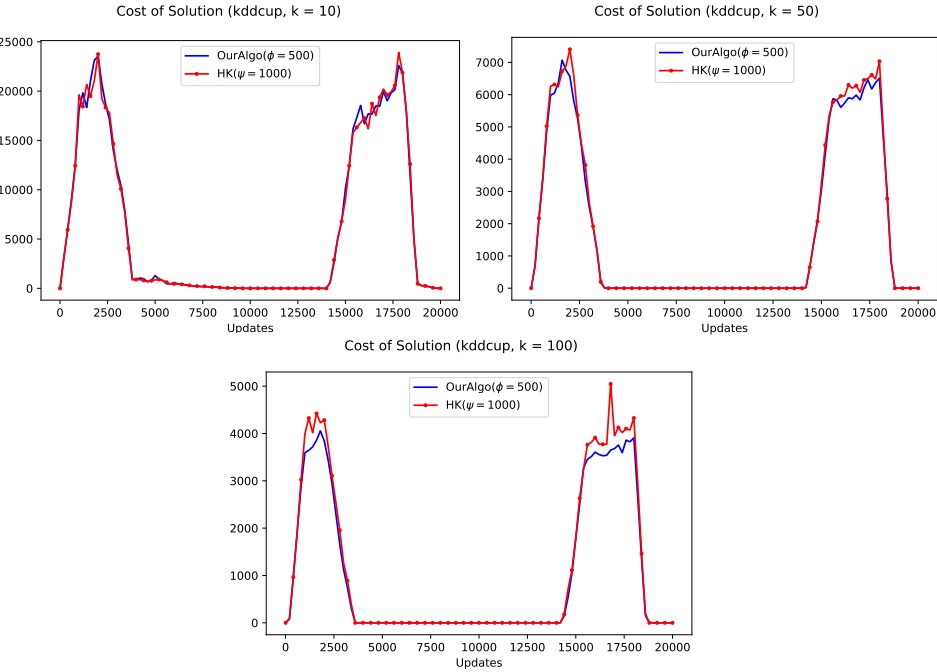

Figure 9: The solution cost by the different algorithms, on KDD-Cup for $k = 10$ (top left), $k = 50$ (top right), $k = 100$ (bottom).

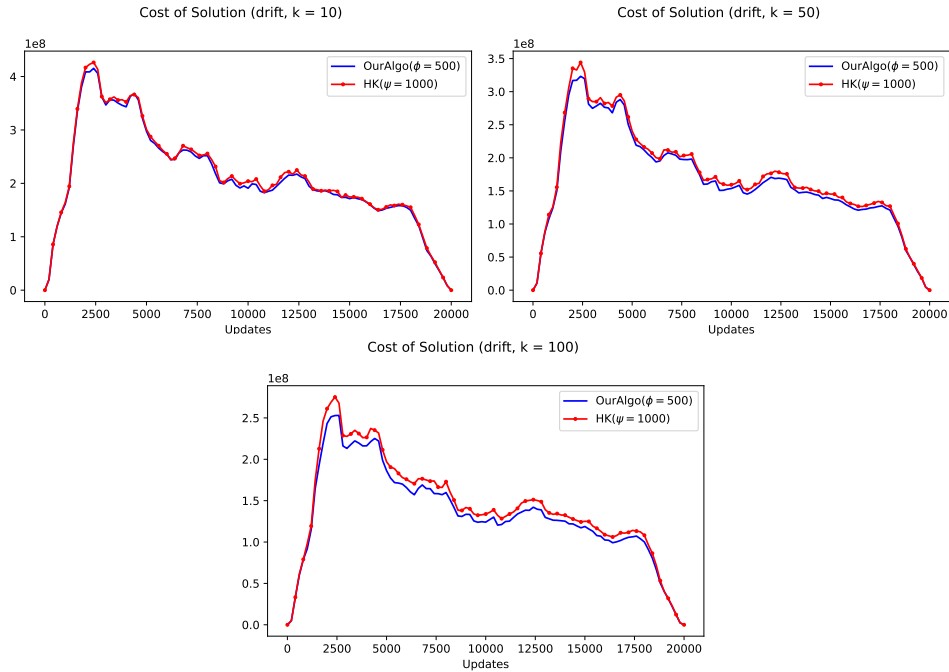

Figure 10: The solution cost by the different algorithms, on Drift for $k = 10$ (top left), $k = 50$ (top right), $k = 100$ (bottom).

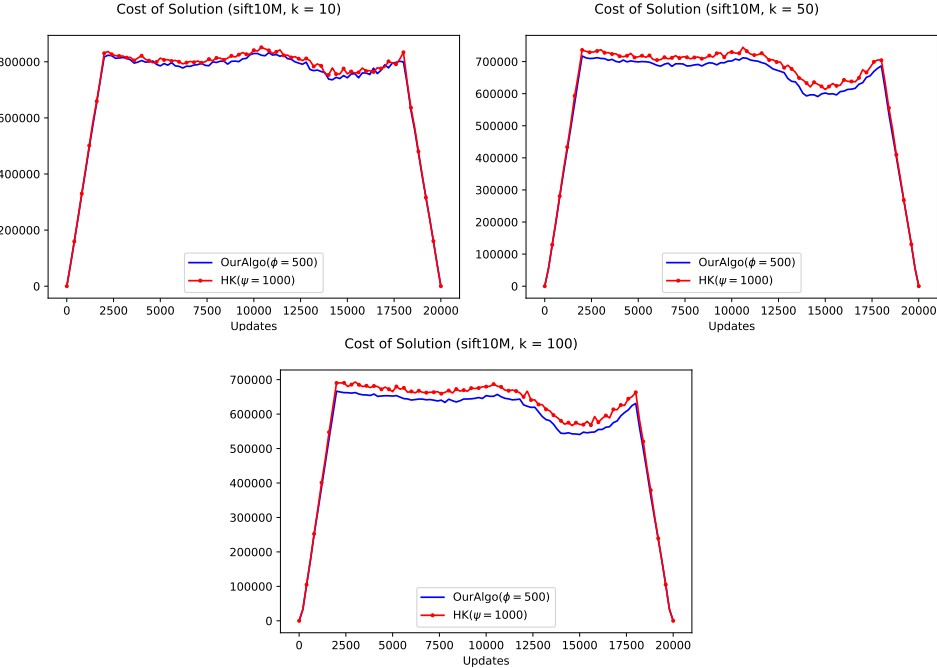

Figure 11: The solution cost by the different algorithms, on SIFT10M for $k = 10$ (top left), $k = 50$ (top right), $k = 100$ (bottom).

### E.3 Query Time Evaluation

Table 4: The average query times for the algorithm OURALG($\phi = 500$) and HK($\psi = 1000$) (we omit the parameter value from the table to simplify the presentation), on the different datasets that we consider and for $k \in \{10, 50, 100\}$.

|  | Song | | Census | | KDD-Cup | | Drift | | SIFT10M | |
|---|---|---|---|---|---|---|---|---|---|---|
|  | OURALG | HK | OURALG | HK | OURALG | HK | OURALG | HK | OURALG | HK |
| $k = 10$ | 0.569 | 0.327 | 0.478 | 0.280 | 0.069 | 0.176 | 0.729 | 0.421 | 0.732 | 0.419 |
| $k = 50$ | 0.610 | 0.347 | 0.511 | 0.295 | 0.075 | 0.141 | 0.784 | 0.447 | 0.795 | 0.448 |
| $k = 100$ | 0.665 | 0.373 | 0.552 | 0.317 | 0.085 | 0.131 | 0.857 | 0.483 | 0.866 | 0.483 |

### E.4 Parameter Tuning

#### E.4.1 Update Time

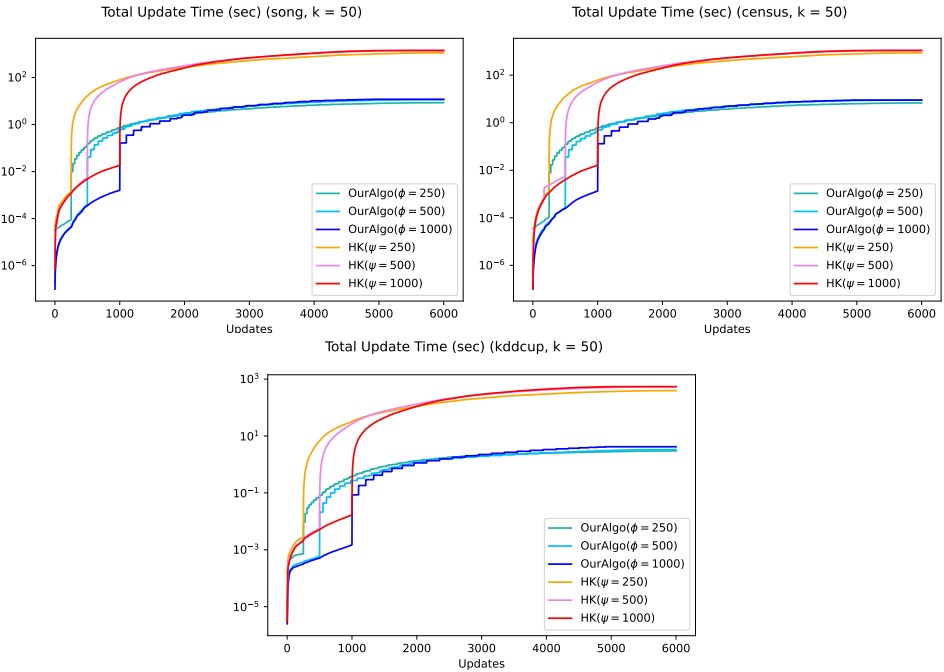

Figure 12: The cumulative update time for different parameters of OURALG and HK, for $k = 50$, on datasets Song (top left), Census (top right), and KDD-Cup (bottom).

### E.4.2 Solution Cost

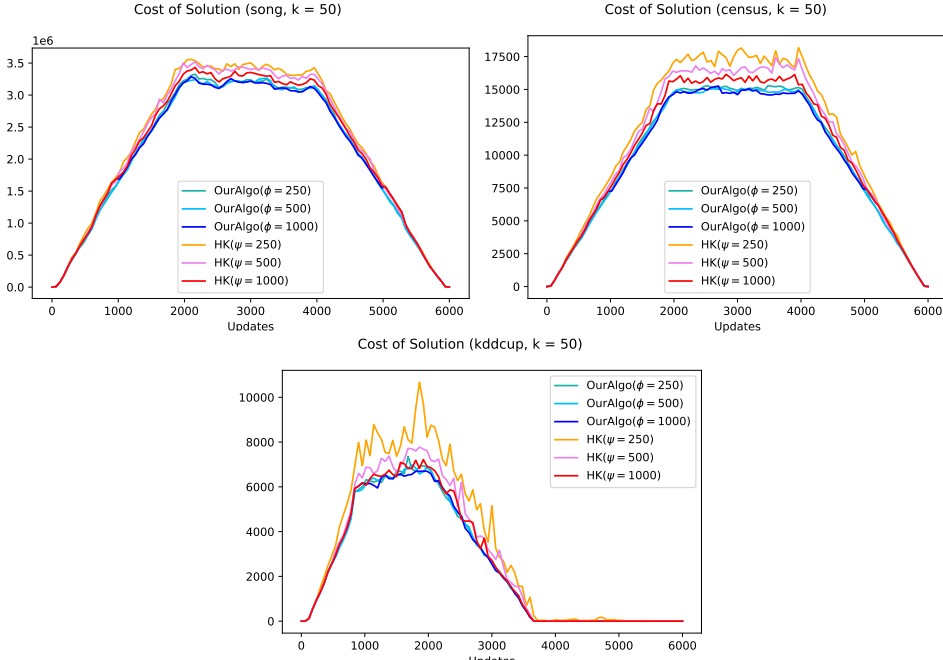

Figure 13: The solution cost for different parameters of OURALG and HK, for $k = 50$, on datasets Song (top left), Census (top right), and KDD-Cup (bottom).

### E.4.3 Query Time

Table 5: The average query times for the algorithm OURALG and HK with different parameters, on the different datasets for $k = 50$.

|  | Song | Census | KDD-Cup |
|---|---|---|---|
| HK($\psi = 250$) | 0.026 | 0.021 | 0.012 |
| HK($\psi = 500$) | 0.087 | 0.073 | 0.043 |
| HK($\psi = 1000$) | 0.293 | 0.249 | 0.156 |
| OURALG($\phi = 250$) | 0.223 | 0.187 | 0.054 |
| OURALG($\phi = 500$) | 0.439 | 0.364 | 0.086 |
| OURALG($\phi = 1000$) | 0.719 | 0.605 | 0.146 |

## E.5 Randomized Order of Updates

### E.5.1 Update Time

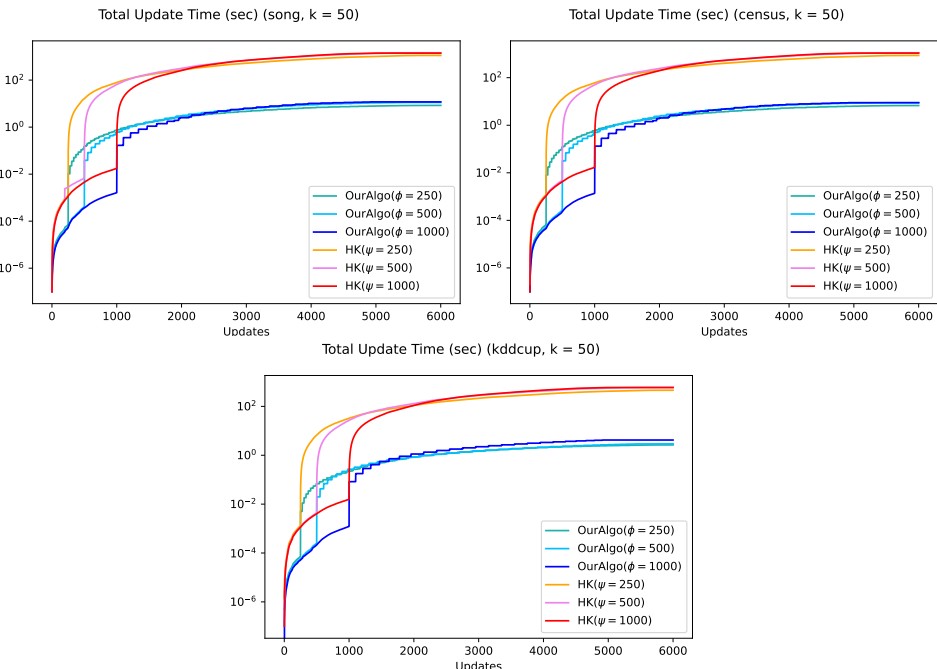

Figure 14: The cumulative update time for different parameters of OURALG and HK, for $k = 50$, over a sequence of updates given by a randomized order of the points in the dataset, on the datasets Song (top left), Census (top right), and KDD-Cup (bottom).

### E.5.2 Solution Cost

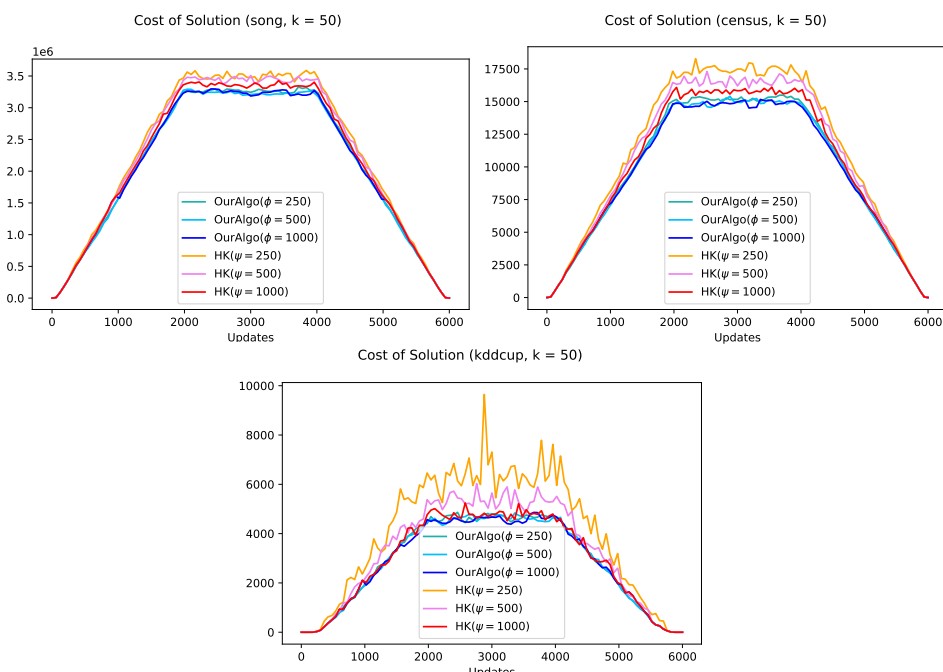

Figure 15: The solution cost for different parameters of OURALG and HK, for $k = 50$, over a sequence of updates given by a randomized order of the points in the dataset, on the datasets Song (top left), Census (top right), and KDD-Cup (bottom).

### E.5.3 Query Time

Table 6: The average query times for the algorithm OURALG and HK with different parameters, for $k = 50$, over a sequence of updates given by a randomized order of the points in each of the datasets that we consider.

|                          | Song  | Census | KDD-Cup |
|--------------------------|-------|--------|---------|
| HK($\psi = 250$)         | 0.025 | 0.021  | 0.014   |
| HK($\psi = 500$)         | 0.086 | 0.073  | 0.050   |
| HK($\psi = 1000$)        | 0.292 | 0.247  | 0.173   |
| OURALG($\phi = 250$)     | 0.225 | 0.185  | 0.062   |
| OURALG($\phi = 500$)     | 0.440 | 0.364  | 0.100   |
| OURALG($\phi = 1000$)    | 0.723 | 0.605  | 0.165   |

### E.6 Larger Experiment

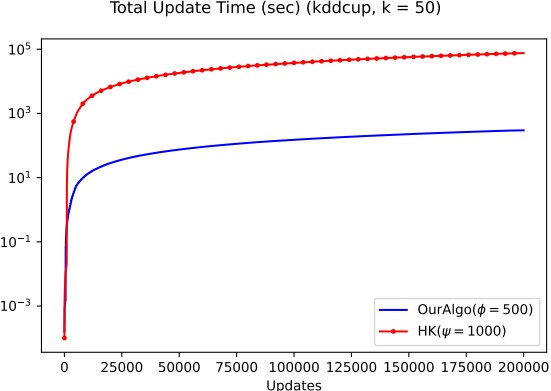

Figure 16: The total update time for OURALG($\phi = 500$) and HK($\psi = 1000$), on the larger instance derived from KDD-Cup, for $k = 50$.

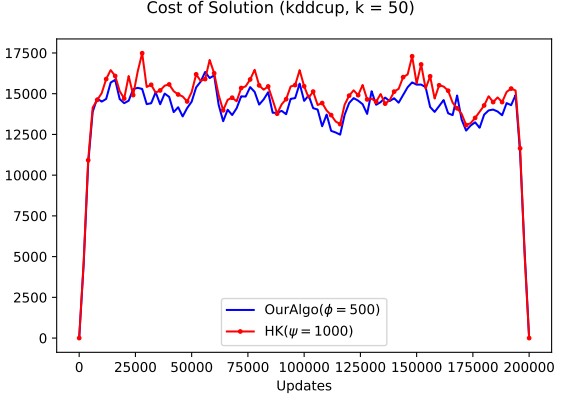

Figure 17: The solution cost produced by OURALG($\phi = 500$) and HK($\psi = 1000$) two algorithms, on the larger instance derived from KDD-Cup, for $k = 50$.

The average query times for OURALG($\phi = 500$) and HK($\psi = 1000$) while handling this longer sequence of updates were $0.416$ and $0.225$ respectively.

