# OpenReview forum: "Fully Dynamic $k$-Clustering in $\tilde O(k)$ Update Time"
_NeurIPS.cc/2023/Conference — NeurIPS 2023 poster_

### Official Review · Reviewer_f4hU · 2023-07-02

**Soundness:** 3 good
**Presentation:** 3 good
**Contribution:** 3 good
**Rating:** 6
**Confidence:** 3

**Summary:**

This paper studies dynamic algorithms for k-median (and k-means) in general metrics. The main result is an \tilde{O}(k) update time, poly(k) query time algorithm that achieves O(1)-approximation. Here, the update model is point insertion/deletion, and the algorithm has access to the distance oracle. A query operation asks to return a list of k points that is O(1)-approximate.

The result improves over the k^2 update time achieved in HK20. This is achieved using a somewhat different but more direct approach. In particular, in HK20, a general reduction to coreset via merge-and-reduce framework was employed, but this work employs a direct dynamization of MP04.

Comprehensive experiments are also provided. Compared with HK20 as a baseline, the new algorithm runs faster and achieves an overall better accuracy.

**Strengths:**

- The paper achieves near-linear in k update time, which is a nice improvement over the previous k^2
- The experiments are convincing, and justify the theoretical improvement
- A nearly-matching lower bound is provided

**Weaknesses:**

- This work only yields O(1)-approx, while HK20 uses coreset approach and hence can build an eps-coreset even for general metrics (even though it does not imply an efficient algorithm for finding a solution). This also means this work cannot be easily generalized to the Euclidean case, where near-linear time PTAS was known.
- The query time is k^2, which may be improved

**Questions:**

- The first paragraph of "our techniques" tried to explain why HK20 has k^2 dependence, but it's not very clear. Here, the black-box coreset is linear in k, and the depth of the binary tree is O(log n). Then how come we have another k factor?
- Followup of the above question: is it true that HK20 relies on a bi-criteria solution? Is that part constitute the bottleneck? Also, does HK20 give bi-criteria solution whose update time is near-linear in k? If so, then maybe it's also an interesting angle to compare with their result, since yours may be viewed as a (strict) improvement of it.
- In the end of Sec 1, it is mentioned that an algorithm for weighted k-median is presented in Sec 2. But I didn't find it
- In Sec 3.1, it seems the algorithm only returns a bi-criteria solution. What's the procedure for making it a real feasible solution? How does that procedure affects the final time complexity?

**Limitations:**

I didn't find these explicitly discussed. A discussion of limitations, and possibly mentioning future directions, could be helpful. This paper is a theory paper and I don't see a potential negative societal impact.

---

> ### Author Rebuttal · Authors · 2023-08-08
>
> We thank the reviewer for their effort and the review.
>
> Weakness 1: We would like to point out here that in the HK20 result, if one wishes to ensure that $\epsilon$ is small, then the resulting coreset has to be very large. For example, for the HK20 guarantee to hold with probability $½$ and $\epsilon=1$, the size of the output coreset would need to be at least $3.3$ million points in size, when $n=2000$ and $k=100$. So in many real world scenarios, the HK20 algorithm would need a coreset that contains the full dataset, to have the concerned theoretical guarantees.
>
> Weakness 2: The ultimate goal in this line of research is to obtain a $O(1)$ approximate dynamic $k$-median algorithm with $\tilde{O}(k)$ update time and $\tilde{O}(k)$ query time. We made substantial progress towards this goal by bringing down the update time from $\tilde{O}(k^2)$ to $\tilde{O}(k)$. Bringing the query time down to $\tilde{O}(k)$ remains a tantalising open question.
>
> Question 1: While the sizes of the corsets is $\tilde{O}(k)$, the time taken to compute these coresets (on inputs of size $\tilde{O}(k)$) is $\tilde{O}(k^2)$, which is why the update time is $\tilde{O}(k^2)$. This is because HK20 needs to recompute the coresets after each update, to ensure that the output is a valid coreset at all times.
>
> Question 2: Our focus on this paper has been to consider the original $k$-median problem, without any bi-criteria approximation. But it is true that we can also maintain a bicriteria approximation in $\tilde{O}(k)$ update time.
>
> Also, it is indeed the case that the bottleneck in the static coreset construction used by HK20 is the computation of a bi-criteria solution (while the rest of the coreset construction can be done in $\tilde{O}(k)$ time), which is the reason why HK20 takes $\tilde{\Omega}(k^2)$ time to handle an update. While the output coreset of HK20 can also be viewed as a bi-criteria solution, it’s still not clear how this can be maintained in $o(k^2)$ update time using the HK20 framework.
>
> Question 3: We apologise for the confusion. We actually intended to cite the definition of the weighted k-median problem, not the algorithm. We use a weighted k-median algorithm as a black box, as described in the proof of Corollary 3.5. We will clarify this point in the final version.
>
> Question 4: This is answered in line 177 (towards the end of Section 3.1). In the final version of the paper, we will take this comment into account and make appropriate changes.
>
> Limitations: Thanks for pointing this out. In the final version of the paper, we will add a paragraph at the very end, pointing out the key, remaining open problem in this topic - namely, to get a constant approximate dynamic $k$-median algorithm with $\tilde{O}(k)$ update time AND $\tilde{O}(k)$ query time.

---

> > ### Comment · Reviewer_f4hU · 2023-08-10
> >
> > Thanks for the response.
> >
> > I'm fine with most of your comments.
> >
> > But there's one thing I'm not sure: why HK20 with eps = 1 and 0.5 success probability (and n, k as you set) require to use 3.3 million points? How do you derive this number of "3.3 million"? Did you obtain this from some experiment, or just a simple calculation from their worst-case bounds?

---

> > > ### Author Response · Authors · 2023-08-10
> > >
> > > This number follows from the bounds in the theoretical guarantees of the HK algorithm and the bound on the number of points needed by the static coreset construction of Braverman et al. (which is used by HK as a black box) in order to get good approximation.
> > >
> > > Since this coreset approach is not practical if we want to maintain theoretical guarantees (as illustrated by this calculation), in order to give a fair comparison of the algorithms, in our experiments we work in a regime which does not guarantee any worst case bound.

---

> > > > ### Comment · Reviewer_f4hU · 2023-08-18
> > > >
> > > > Thanks for the clarification.
> > > >
> > > > However, it seems we are somehow mixing two ways of evaluating the coreset: the worst-case theoretical bound, and the practical performance.
> > > >
> > > > Theoretically, I still think that your result is not a strict improvement over [HK20], because of the worst approximation ratio. But I also agree that yours could indeed perform better than [HK20] in many practical scenarios, as your experiments explain.
> > > >
> > > > Overall, I think that this is an interesting result. However, the weaknesses that I mention seem to be fair. Thus, I would like to keep my scores unchanged.

---

### Official Review · Reviewer_nK4h · 2023-07-05

**Soundness:** 3 good
**Presentation:** 2 fair
**Contribution:** 2 fair
**Rating:** 5
**Confidence:** 2

**Summary:**

This paper studies the $k$-median/means problems in fully dynamic settings. Clustering in fully dynamic setting is a recent hot topic, where fully dynamic $k$-center has been well studied in literature. However, little was known for fully dynamic $k$-median/means problems. Inspired by the static framework proposed by Mettu and Plaxton, where a minimum radius ball coverage strategy is used to obtain $t=O(logn)$ layers of representations for good approximation of the given $k$-median/means instance. This paper modifies the classic framework by Mettu and Plaxton and presents an $O(1)$-approximation algorithm for the fully dynamic $k$-median/means problems with $\tilde{O}(k)$ amortized update time and $\tilde{O}(k^2)$ worst case query time, which improves the previous coreset-based method with $\tilde{O}(k^2)$ worst case update time.

**Strengths:**

1. This paper proposes simple but efficient approximation algorithm for the fully dynamic $k$-median/means problems with $\tilde{O}(k)$ amortized update time and $\tilde{O}(k^2)$ worst case query time, which improves the previous result with $\tilde{O}(k^2)$ update time and $\tilde{O}(k^2)$ query time. The authors also show that the update time of our algorithm is optimal up to polylogarithmic factors.

2. This paper gives detailed experimental evaluation of fully dynamic k-median algorithms for general metrics and shows that the proposed framework is more efficient than previous ones.

**Weaknesses:**

1. The techniques used in this paper seem to rely heavily on the minimum ball coverage method by Mettu and Plaxton.

2. The challenges for obtaining good update time and query time is not well discussed.

3. The theoretical analysis is not an easy read in a limited time. The intuition behind the analysis and algorithm before going into the details of lemmas and proofs should be given before the proofs.

**Questions:**

1. In each insertion, why should the data point to be inserted added to each layer $i \in [t]$ instead of one of the $t$ layers?

2. Can the authors discuss the challenges for applying the Mettu's and Plaxton's method for solving the fully dynamic $k$-clustering problems?





**Limitations:**

Since this is a theoretical paper, I don't think there is potential negative societal impact of this paper.

---

> ### Author Rebuttal · Authors · 2023-08-08
>
> We thank the reviewer for their effort and the review.
>
> Regarding the challenges behind getting our result: Typically, the first challenge towards designing a dynamic algorithm for a given problem is to identify a suitable static algorithm which is flexible enough that it can be adapted to the dynamic setting. This is emphasised in line 83 (and the paragraph preceding it), where we explain that the static algorithm that we build upon is completely different from prior work. Next, even after identifying the Mettu-Plaxton algorithm as our starting point, we have to note that simply following a strategy of corresponding to the output of the static algorithm at every time-step will lead to very large update time. To circumvent this difficulty, we have to suitably modify the Mettu-Plaxton algorithm so that it lazily waits until it accumulates sufficiently many updates at some layer $j$, and then reconstructs all the layers $i \geq j$. Thus, we need to be lazy in a ``fine-grained’’ manner. We cannot simply say that we wait for a certain period of time, and then run the static algorithm again from scratch on the whole input. We will clarify this point in the final version.
>
> Regarding handling an insertion: Note that the layers are nested within one another (see line 4 of Algorithm 1). Thus, if we add the point $x$ being inserted to layer $U_i$, then we must add it to all layers $j \leq i$. Now, one might ask why can’t we just add the point $x$ only to $U_1$ and be done with it? The reason is this: Then necessarily $x$ will become part of the set $C_1$ (see line 4 of Algorithm 1), but there will be no guarantee that this point $x$ in $C_1$ is covered by some ball of radius $\nu_1$ around $S_1$ (see lines 124 - 131).
>
> Regarding the presentation in the paper: In Section 3.1, we described the static algorithm by Mettu-Plaxton, which was our starting point. In Section 3.2, we provided some intuitions regarding the modifications we need to make in order to make it dynamic (see lines 169 - 172), and along with it described concretely the dynamic algorithm. For space constraints, we had to defer the complete proofs for the analysis to the appendix.

---

> > ### Comment · Reviewer_nK4h · 2023-08-19
> >
> > Thank you for the clarification. I keep my evaluation of the paper unchanged.

---

### Official Review · Reviewer_REqo · 2023-07-06

**Soundness:** 4 excellent
**Presentation:** 3 good
**Contribution:** 4 excellent
**Rating:** 7
**Confidence:** 4

**Summary:**

Fully dynamic k-clustering in O(k) update time

This paper studies fully dynamic k-clustering. It gives a fully dynamic algorithm that maintains O(1)-approximate solutions to k-median and k-means with \tilde O(k) amortized update time and \tilde O(k^2) worst-case query time. On the negative side, the authors showed that the Omega(k) amortized update time is required if one needs to achieve O(1)-approximation ratio and poly(k) query time. So they gave the optimal update time, and the time improves the prior best-known update time of O(k^2).  The authors also did experiments to complement the theoretical analysis.

The dynamic algorithm is built on the static algorithm of [27]. It runs for many iterations. In each iteration i, the algorithm samples a set S_i of points and creates a set of smallest-radius balls around the samples so that the balls cover at least beta fraction of the remaining points, for a constant beta. The algorithm then removes the covered points and repeats the process.  This gives a partition of the points into \tilde O(k) balls. They construct an assignment that maps every point to its ball center. Then a k-median or k-means solution can be constructed using the centers only.

The dynamic algorithm maintains the tree-structure constructed in the static algorithm in a lazy manner by allowing some slack at many places. It needs to rebuild the tree from some layer if the cost of the assignment at that layer or above becomes bad. On average, it needs many updates for the algorithm to rebuild a tree from some layer, giving a good amortized update time.



**Strengths:**

Overall, the paper gives a tight update time of \tilde O(k) for the fully dynamic k-clustering problem, using elegant techniques. This is a solid accept.

**Weaknesses:**

The hidden approximation ratio is a little big.

**Questions:**

1. Can you give a rough bound on the approximation ratio of the algorithm?

2. What happens if each update contains a batch of p points? Can you achieve an update time of O(p + k), instead of O(pk)?

**Limitations:**

No limitations.

---

> ### Author Rebuttal · Authors · 2023-08-08
>
> We thank the reviewer for their effort and the review.
>
> Q1: The approximation ratio of our algorithm is (very close to) a factor $4$ off the approximation ratio of the static algorithm by Mettu-Plaxton. This is about $83 + 168\alpha$, where $\alpha$ is the approximation ratio of the static algorithm used to handle queries. However, we observe in our experiments that the quality of the solution returned by our algorithm is significantly better than the one predicted by the analysis.
>
> Q2: There is an existing lower bound which asserts that for any $k$, a constant approximate k-median algorithm takes $\Omega(nk)$ time in the static setting. If our dynamic algorithm could handle an update consisting of a batch of $p$ points in $o(pk)$ time, then this would imply the existence of a static algorithm that runs in $o(nk) + O(k^2)$ time (by passing all the n points to the dynamic algorithm in one batch and making a query). In the event that $k = o(n)$, this leads to a contradiction. Hence, any constant approximate dynamic algorithm must take $\Omega(pk)$ time to handle a batch update consisting of $p$ points (as long as the query time is polynomial in $k$).

---

> > ### Comment · Reviewer_REqo · 2023-08-18
> >
> > Thank the authors for the responses. I have no more questions.

---

### Official Review · Reviewer_wnsQ · 2023-07-07

**Soundness:** 4 excellent
**Presentation:** 4 excellent
**Contribution:** 3 good
**Rating:** 6
**Confidence:** 4

**Summary:**

This paper considers the dynamic version of the $k$-median and the $k$-means clustering problem in an arbitrary metric space. The authors provide an $O(k)$ amortized time (which is near optimal based on a lower bound that the authors provide) for insertions and deletions of points and a query time of $O(k^2)$. This is an improvement of the recent result by Henzinger and Kale, ESA 2020 which provided a dynamic algorithm with a worst case $O(k^2)$ update time. The algorithm in this paper is based on making the algorithm of Mettu and Plaxton dynamic. The authors also provide implementations of their algorithm as well as prior works (including a coreset construction algorithm) and then provide an empirical performance analysis of these algorithms.

**Strengths:**

This paper is providing a near-optimal bound for amortized update time for dynamic k-means and k-median clustering problems.

 The authors provide implementations of their algorithm as well as prior works which is very useful.


**Weaknesses:**

In terms of techniques, the methods used are adaptation of an existing algorithm of Mettu and Plaxton and perhaps somewhat incremental in nature.

**Questions:**

My questions were answered in a previous review cycle

---

> ### Author Rebuttal · Authors · 2023-08-08
>
> We thank the reviewer for their effort and the review.
>
> We agree that we extend the Mettu-Plaxton algorithm to the dynamic setting. We believe this to be an important contribution, since our algorithm is simple and practical to implement, and improves upon the prior bound for this fundamental clustering problem in the dynamic setting.

---

### Decision · Program_Chairs · 2023-09-21

**Decision:**

Accept (poster)

**Comment:**

The paper is generally supported by the reviewers.

The strength of the submission is in improving the update time of an important dynamic clustering problem and the reviewers also appreciate the improved performance in many scenarios in the experiments. The new algorithm generally improves on the previous dynamic algorithm in both time and solution quality.

On the downside, the techniques are somewhat incremental in nature as it is a modification of an existing static algorithm. The theoretical approximation factor is very high compared with previous works so even though it performs well in the scenarios in the experiments, there could be other scenarios where it does not improve the previous works.